# Tune out pain: Agency and active engagement predict decreases in pain intensity after music listening

**Claire Howlin**[1]*, **Alison Stapleton**[2], **Brendan Rooney**[2]

**1** Department of Biological and Experimental Psychology School of Biological and Experimental Psychology, Queen Mary University of London, London, United Kingdom, **2** School of Psychology, University College Dublin, Belfield, Dublin, Ireland

* claire.howlin@qmul.ac.uk

**Data Availability Statement:** All anonymous data are shared publicly. Data are available on the Open Science Framework. https://osf.io/4ywjd/ DOI 10.17605/OSF.IO/4YWJD.

**Funding:** This research was supported by Nurofen funding (https://www.nurofen.co.uk/) awarded to

## Abstract

Music is increasingly being recognised as an adjuvant treatment for pain management. Music can help to decrease the experience of both chronic and experimental pain. Cognitive agency has been identified as a specific mechanism that may mediate the analgesic benefits of music engagement however, it is unclear if this specific mechanism translates to acute pain. Previous attempts to understand the cognitive mechanisms that underpin music analgesia have been predominantly lab-based, limiting the extent to which observed effects may apply to participants' everyday lives. Addressing these gaps, in naturalistic settings, the present study examined the degree to which cognitive agency (i.e., perceived choice in music), music features (i.e., complexity), and individual levels of musical sophistication were related to perceived pain. In an online global experiment, using a randomised between groups experimental design with two levels for choice (no choice and perceived choice) and two levels for music (high and low complexity), a sample of 286 adults experiencing acute pain reported their pain intensity and pain unpleasantness pre- and post-music listening. A bespoke piece of music was co-created with a commercial artist to enable the manipulation of music complexity while controlling for familiarity, while facilitating an authentic music listening experience. Overall, findings demonstrated that increased perceived control over music is associated with analgesic benefits, and that perceived choice is more important than music complexity. Highlighting the importance of listener engagement, people who reported higher levels of active engagement experienced greater decreases of pain intensity in the perceived choice condition, than those who reported lower levels of active engagement. These findings have implications for both research and practice, emphasising the importance of facilitating freedom of choice, and sustained engagement with music throughout music listening interventions.

CH (Ref.No.: 70037). The funders had no role in study design, data collection and analysis, decision to publish, or preparation of the manuscript.

**Competing interests:** The authors have declared that no competing interests exist.

# Introduction

Music listening is increasingly being recognised as an adjuvant treatment for pain management [1, 2]. But there is still an open question in terms of what is driving the analgesic benefits of music listening. Previous studies that have targeted basic acoustic features such as tempo [3, 4], energy [5], or combinations of features leading to perceived relaxing properties in the music [6] or perceived arousal, valence and depth [7]. Yet these studies have found no relationship between basic perceptual properties, and decreases in pain perception. Indeed, the evidence from meta-analyses supports the view that specific features do not predict reductions in pain, which can be measured in terms of pain intensity and pain unpleasantness [4, 8]. Instead, it has been observed that effective music interventions can vary widely in terms of genre, tempo, loudness, duration, timing, and equipment used [3, 4, 9]; yet still achieve comparable results. This precludes the notion that a one-to-one relationship exists between musical features and physiological responses, and emphasises the role of *functional equivalence* in music interventions. This means that not only do different people often respond very differently to the same piece of music, reciprocally people can use different pieces of music to get the same functional effect of analgesia [10].

Given the lack of evidence to support the role of basic acoustic features in music listening interventions, and as predicted by the cognitive vitality model for music interventions for pain [11], we argue that greater consideration should be given to higher level cognitive processes, such as (i) cognitive agency (e.g. self-directed choice) and (ii) active cognitive engagement with music (e.g. active listening, musical absorption, self-reflection) [11, 12]. The cognitive vitality model outlines that a meaningful, rewarding, absorbing musical experience enhances the analgesic benefits of music [11]. The potential reward of music listening acts as an incentive for the person to continue to listen and maintain active engagement. Prolonged active engagement with music can lead to *musical absorption*, which is related to a decreased awareness of physical sensations, and the initial pain becomes less salient. Additionally, the cognitive vitality model outlines that cognitive agency can enhance active engagement with music as people decide to listen more closely to specific streams in the music or focus on the lyrics.

Cognitive agency refers to an individual's sense of control over their environment and contributes to greater feelings of health and wellbeing [5, 11, 13, 14]. The impact of cognitive agency in music interventions is underpinned by neuroimaging research that implicates the role of the default mode network in mediating the analgesic responses to pain [15–17]. This emphasizes that music is involved in a top-down pain regulation strategy that may be enhanced by the act of choosing the music itself. This argument is supported by studies [2, 7, 16] and meta-analyses [8, 9] that demonstrate that patient preferred, self-chosen music is an important predictor of a successful music intervention for pain across a range of clinical and experimental contexts. Previous studies have demonstrated that increased agency over musical production can lead to a decrease in perceived physical effort during physical exercise [17]. A potential limitation of empirical work exploring this is that the act of making a choice of music is confounded by other things such as participants' personal connection to the music or familiarity.

A recent experiment used a novel perceived choice paradigm to isolate the impact of cognitive agency on pain. In this study participants were given different degrees of perceived control over music selection, when in fact the music was predetermined by the experimenter [5]. The cold pressor task was used to stimulate pain, and when participants believed that they were selecting the music, their pain tolerance increased compared to when they had no control over the music. This experiment indicates that the actual act of making a choice over music can contribute to increases in pain tolerance when the music itself was controlled. The current study aims to extend these findings of acute pain tolerance beyond the lab to participants' pre-

existing everyday acute pain experience. While increasing the ecological validity of the previous study, the current study also aims to increase experimental control by working with a commercial music artist and producer to design a bespoke track specifically composed for this study. This is a methodological step forward compared to lab-based studies. Previous attempts to examine musical engagement in lab settings have used focused task paradigms with tone sequences rather than real pieces of music, which undermines the likelihood that sustained attention could occur and that people will actually enjoy the music [16].

Working with a music artists also allowed us to design a test of the contribution of the music features to pain experience. Individual responses to music are highly idiosyncratic, meaning different features are likely to evoke different types of emotional responses in different individuals [18–21]. Therefore, music analgesia does not arise as an automated or induced response to music [5, 7, 22]. We suggest that music analgesia arises as from an *interaction* between the music and the listener that is characterised by an optimal level of active music engagement [2, 5, 7, 9]. Optimal engagement arises from a music listening experience that is neither too complex nor too simple for the listener [23–25], based on their own personal music preferences, expertise, and baseline state of arousal [26]. This means that the music that a person will find optimal, will vary across different people, and vary for the same person at different times. If the music is too simple for the listener it can lead to boredom, and if it is too complex it can lead to irritation or over-stimulation [26]. Accordingly, the present study will manipulate the complexity of the music.

As mentioned, music engagement is an individual experience and there are a number of individual characteristics that influence the way that people engage with music, including one's musical sophistication, the degree to which people experience musical reward, and an individual's tendency to empathise with the emotional content in music. Musical Sophistication is an individual trait that can account for different levels of musical skills and interest across the general population [27]. Musical reward refers to the neural reward processes that can occur in response to listening to music [28], which tends to occur more frequently when listening to your own self-chosen music [29]. An individual's tendency to empathise with the emotions expressed in a piece of music can be predicted by scores on the interpersonal reactivity index, which also accounts for why some people find enjoyment and pleasure in sad music [18]. Each of these individual traits can be measured using reliable psychometric measures which are used in the current study [27, 30, 31]. It is reasonable to expect that people with higher scores on musicality, musical reward or the interpersonal reactivity index will pre-dispose them to becoming more engaged, distracted, and absorbed by music. Conversely, people with less interest, experience, or who find less enjoyment from music are less likely to engage with it to the same degree.

Also it is important to note that because pain is multidimensional with both physical and emotional components [32], it is important to evaluate both the intensity and unpleasantness associated with a pain experience. Due to the inherently subjective nature of pain, it is not possible to measure pain with an objective measure and so self-report measures are used in pain management research [33]. Numeric rating scales (NRS) are considered the gold standard for measuring patient's subjective feeling of pain intensity and pain unpleasantness, because they are more sensitive than other self-report measures that treat pain as a unidimensional construct [33]. Accordingly, this study uses two measures of pain experience, pain intensity, and pain unpleasantness.

## Research questions and hypotheses

Before collecting data, we pre-registered our hypotheses based on the above rationale (https://osf.io/egqaz). The present study will examine the degree to which perceived control of music is

related to a decrease in perceived pain in the context of everyday acute pain. Here we predict that increased perceived control predicts decreases in pain intensity and pain unpleasantness (H1). Although the benefits of music engagement have been well demonstrated for chronic pain [9, 16, 34], and experimental pain [2, 5, 35], it is not yet known the degree to which the analgesic benefits of music engagement translates to acute pain in everyday settings. Additionally, this study will explore the role of music engagement by manipulating musical complexity via an altered piece of bespoke music specifically composed for this study, to maximise engagement. We also recognise that individual attributes related to musicality (specifically active music engagement) may contribute to the analgesic responses to music for acute pain. Here we predict that pain experience will be different between the low music complexity and high music complexity conditions (H2) we make no specific directional prediction of how this might interact with participants' level of active engagement. Before testing these hypotheses, as a manipulation check, it is important to make sure that the two music tracks used are comparable in terms of aesthetic and emotional responses. To this end, we examine if the tracks are different in terms of aesthetic or emotional responses. While much evidence has already demonstrated the benefits of music listening for pain, testing this is an important pre-requisite to our main hypotheses, given the bespoke nature of the music used in the present study. Finally, because the study recruits a large online sample, we collect data on a range of important individual characteristics listed above (musical sophistication, musical reward, empathy) so as to profile the participants on these important characteristics. In addition, they serve to compare the independent groups in terms of these variables to ensure their comparability.

## Methods

### Experimental design

This study design and analysis were preregistered on the Open Science Framework: https://osf.io/4ywjd/. The present study employed a randomised 2x2 between groups experimental design with two levels for choice (no choice and perceived choice) and two levels for music (high complexity and low complexity). One of the core aims of this study was to replicate a finding that was previously demonstrated in lab setting, to a real world clinical sample with acute pain. Due to the relatively fleeting nature of acute pain, online data collection was used to facilitate more rapid data collection from people at a time when they experienced acute pain.

### Ethics

Participants provided informed written consent and could withdraw at any time. Since the study was classified as a low-risk study, a research ethics exemption was obtained from University College Dublin Human Research Ethics Committee (REFRN: HS-E-21-96). The Human Research Ethics Committee Guidelines and Policies specifically Relating to Research Involving Human Subjects were abided by in all aspects of conducting this research. No personal identifiable data was collected as part of this study, and anonymous links were used to share the study. With the consent of the participants, anonymous data will be retained indefinitely and shared on an open repository to facilitate the principles of open data sharing and open science.

**Sample size.** An a priori power analysis using G*Power indicated that a sample size of 329 participants would be required for a 2 x 2 ANCOVA analysis with 6 co-variates based on an effect size of f = 0.23 in line with the analgesic effect of music identified in a previous meta-analysis [4]. 585 participants completed the online experiment, and 286 participants were found to meet all of the inclusion criteria below, and completed all components of the experiment. A second power analysis was completed based on the observed power (Cohen's $d$ = .73

for pain intensity and $d = .72$ for pain unpleasantness), and so data collection was stopped based on the observed power calculation for a linear model.

## Inclusion criteria

A two-stage screening process was used to identify eligible participants using the Prolific participant recruitment platform. The screening process and inclusion criteria were registered in the study protocol on the open science framework. In stage one 2691 people answered screening questions which were presented one at a time, based on each previous response along with red herring questions to reduce the likelihood that participants would guess the nature of the study. Eligible participants were provided with a link to the full study and invited to take part on later on the same day. A total of 585 participants completed the experiment. Participants were considered eligible for inclusion if they: (i) reported an age over 18 years; (ii) reported baseline pain of at least 2 on a pain intensity numeric rating scale (NRS) ranging from 1 to 10 (to ensure that they were experiencing a mild level of pain before music listening); (iii) experienced pain for less than 12 weeks (pain extending beyond 12 weeks is often classified as chronic pain [36]); (iv) had not taken pain medication in the past 8 hours; (v) were not on any routine prescribed medication; (vi) were not pregnant; (vii) passed all four attention checks during the online experiment; and (viii) had access to headphones. Based on this criteria, 299 participants were excluded because they did not meet the inclusion criteria. The final sample size comprised 286 adults distributed across Europe (38.4%) North America (38.4%), South Africa (11.2%), Australia and New Zealand (2%), Israel (.7%), and Chile (.3%). Median pain duration was between 1 day and 1 week. The most commonly reported type of pain was back pain (34.6%), followed by headache (16.4%), pain in the joints (15%), neck pain (9.6%), and period pain (9.6%) See Table 1 for participants' demographic details and well-being scores. Participants were paid at a rate of £9 per hour for their participation, with an average completion time of 30 minutes.

## Procedure and materials

Participants completed the experiment online through Qualtrics (Qualtrics, Provo, UT). First, participants reported their demographics, well-being, and pre-music listening pain intensity and unpleasantness. Next, using the Qualtrics randomiser participants were randomly allocated to one of the four conditions: (i) no choice and low complexity track, (ii) no choice & high complexity track, (iii) perceived choice and low complexity track, (iv) perceived choice and high complexity track. The perceived music choice paradigm [10] was used to examine the effect of perceived control on pain. In perceived choice conditions, participants 'chose' which track they would like to hear in full by sampling and subsequently selecting one track from four 2-second music clips. The instruction given to participants was '*Listen to each music sample and then choose the music you think would be the best thing to listen to when you have pain*'. Participants in these conditions were unaware that each music clip came from the same piece of music (i.e., each clip was a different part of the same track) and that their final 'chosen' song was predetermined by their random condition assignment. All participants listened to their assigned piece of music in its entirety. Once the music finished, participants reported their (i) post-music listening pain intensity and unpleasantness, (ii) ratings of their emotional responses to the music, and (iii) individual attributes related to musicality, trait empathy, and musical anhedonia (see Fig 1).

## Pain outcomes—Pain intensity and unpleasantness

Participants rated their pain intensity and unpleasantness before listening to the music on Numeric Rating Scales (NRS). The 100-point intensity scale had three anchor points 'no pain'

**Table 1. Participant demographics and health and wellbeing scores for each experimental condition.**

|  | 1 | 2 | 3 | 4 |  |  |
| --- | --- | --- | --- | --- | --- | --- |
| Experimental Condition | No Choice Low Complexity (n = 73) | No Choice High Complexity (n = 70) | Perceived Choice Low Complexity (n = 67) | Perceived Choice High Complexity (n = 76) | Total (n = 286) |  |
|  | M (SD) | M (SD) | M (SD) | M (SD) | M (SD) | α |
| Age | 35.44(12.74) | 34.29(13.53) | 32.83(13.63) | 34.68(13.47) | 34.35(13.31) |  |
| Gender (Frequency) |  |  |  |  |  |  |
| Female | 38 | 42 | 40 | 44 | 164 |  |
| Male | 33 | 27 | 27 | 32 | 119 |  |
| Self-Described | 2 | 1 | 0 | 0 | 3 |  |
| Musical Attributes |  |  |  |  |  |  |
| GMSI F1 Active Engagement[a] | 36.95(7.51) | 36.2(7.77) | 38.93(6.85) | 36.75(6.60) | 37.17(7.23) | .79 |
| GMSI F2 Perceptual Abilities | 41.49(5.13) | 41.47(5.63) | 41.4(5.21) | 41.29(5.32) | 41.41(5.296) | .76 |
| GMSI F3 Musical Training | 21.78(5.42) | 21.2(5.75) | 21.39(5.77) | 21.45(5.98) | 21.46(5.71) | .88 |
| GMSI F4 Emotions | 29.95(4.25) | 29.3(4.33) | 30.04(3.41) | 29.32(4.31) | 29.64(4.10) | .70 |
| GMSI F5 Singing Abilities | 29.7(5.61) | 28.73(6.93) | 29.09(5.75) | 29.21(5.47) | 29.21(5.94) | .81 |
| Barcelona Musical Reward | 79.26(9.32) | 79.29(9.54) | 80.16(8.63) | 79.03(10.05) | 79.42(9.38) | .84 |
| IRI Fantasy Subscale | 16.29(6.89) | 17.56(6.07) | 16.96(7.34) | 16.86(5.98) | 16.91(6.56) | .85 |
| Health and Wellbeing |  |  |  |  |  |  |
| Health Status[b] | 8.32(2.01) | 8.3(1.97) | 8.1(1.93) | 8(1.88) | 8.18(1.94) | .78 |
| Wellbeing[c] | 6.23(2.93) | 5.59(3.12) | 5.43(2.67) | 5.61(2.82) | 5.72(2.89) | .61 |
| Cause of Pain (Frequency) |  |  |  |  |  |  |
| Back Pain | 27 | 18 | 25 | 28 | 98 |  |
| Headache | 17 | 7 | 13 | 9 | 46 |  |
| Joint Pain (e.g. hips) | 7 | 15 | 10 | 12 | 44 |  |
| Neck Pain | 6 | 11 | 5 | 6 | 28 |  |
| Period Pain | 3 | 11 | 7 | 6 | 27 |  |
| Other | 9 | 1 | 1 | 6 | 17 |  |
| Injury | 2 | 3 | 5 | 4 | 14 |  |
| Toothache or Earache | 1 | 2 | 0 | 3 | 6 |  |
| Stomach Ache | 1 | 2 | 1 | 2 | 6 |  |

*Notes.* Abbreviations GMSI Goldsmiths Musical Sophistication Index, IRI Interpersonal Reactivity Index, M Mean, SD Standard Deviation, f frequency. α The within-sample reliability co-efficient for each subscale was calculated using Cronbach's alpha.

[a]Possibly due to coronavirus and its associated lockdowns/ public health restrictions, one item from the GMSI Active Engagement scale ('I don't spend much of my disposable income on music') did not load well onto the scale and was deleted. The scale was re-calculated with 8 items for analysis.

[b]Health status was assessed using the 4-item HowRu Health Status measure [37].

[c]Personal well-being was assessed using the 4-item personal well-being measure [38].

(0), 'moderate pain' [39], and 'worst pain imaginable' (100). The 100-point unpleasantness scale ranged from 'not unpleasant' (0) to 'extremely unpleasant' (100). After participants listened to the music, they provided a second set of pain intensity and pain unpleasantness on identical rating scales.

**Ratings of emotional responses to music.** Ratings of emotional responses to the music were assessed using the 9-item Geneva Emotional Musical Scale (GEMS-9; [40]). Participants

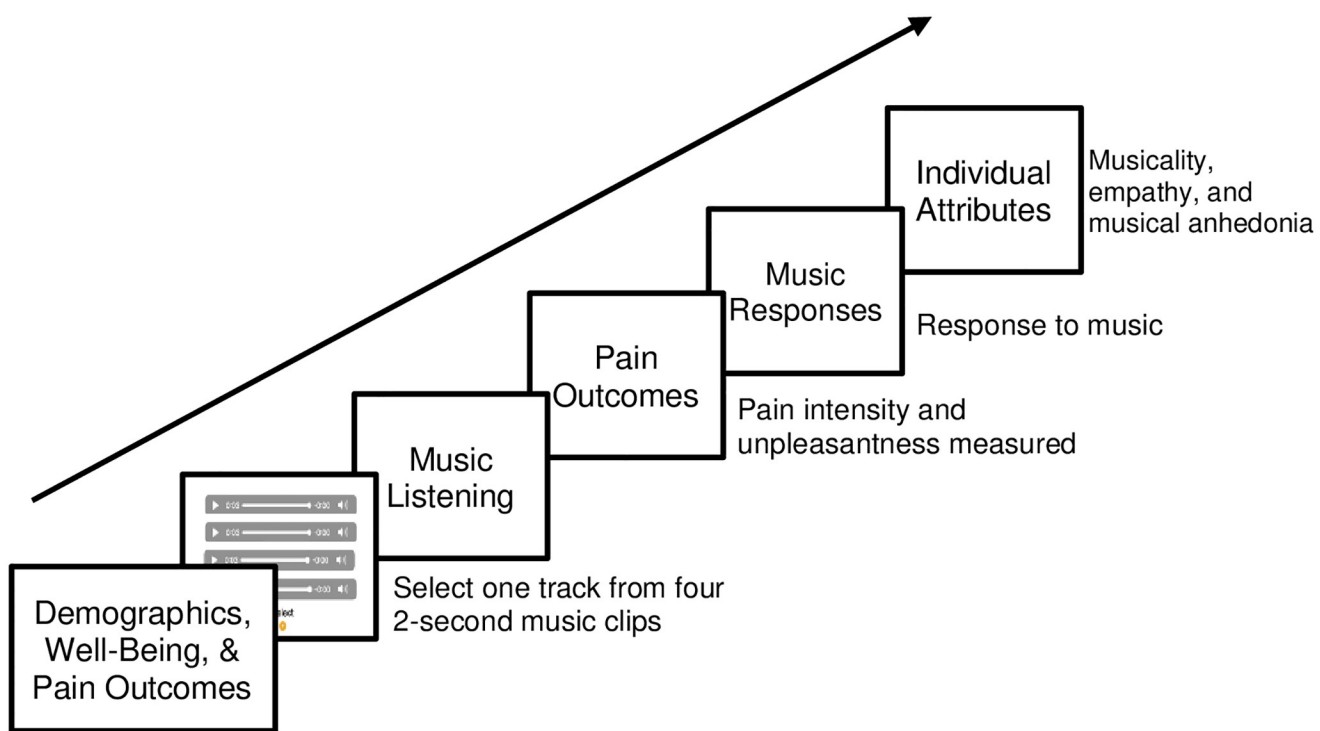

**Fig 1. Experimental procedure.** Participants were randomly allocated to one of four groups: (i) no choice and low complexity track, (ii) no choice and high complexity track, (iii) perceived choice and low complexity track, (iv) perceived choice and high complexity track. In the perceived choice conditions, participants listened to four 2-second music clips and selected which piece of music they wanted to listen to in full. Participants were naïve to the fact that they were listening to different parts of the same song (i.e., the final 'chosen' song was predetermined by their randomly assigned experimental condition.

rated their assigned track across nine emotional dimensions (e.g., Wonder, Transcendence, Power, Tenderness, Nostalgia, Peacefulness, Joyful Activation, Sadness, and Tension) on a five-point Likert scale ranging from 1 ('Not at all') to 5 ('Very much'). Research has supported the factor structure of the GEMS-9, further demonstrating that it provides a better account of emotional responses to music than non-domain-specific emotional models [40].

### Individual traits

**Musical sophistication.** The Goldsmiths Musical Sophistication Index v1.0 (GMSI; [27]) was used to measure musical sophistication, because it has been designed for use with the general population and demonstrates strong test-retest reliability and psychometric validity. The GMSI is comprised of five subscales (active engagement, perceptual abilities, musical training, emotional engagement, and singing abilities), described below that measure an individual's tendency to engage with music on cognitive and emotional dimensions. *Active Engagement* captures a range of active musical engagement behaviours as well as the deliberate allocation of time and money on musical activities. *Perceptual Abilities* represents the self-assessment of a cognitive musical ability and is mostly related to music listening skills. *Musical Training* accounts for the extent of musical training and practice.

*Emotions* accounts for active behaviours related to emotional responses to music. *Singing Abilities* reflects different skills and abilities related to singing. Possible scores on the GMSI range from 18–126, with an average score of 81.58 across the general population. Participants responded on a 7-point Likert scale which ranged from 'completely disagree' (1) to 'completely

agree' [6]. Population studies support the internal consistency, test-retest reliability, and validity of the GMSI [27]).

**Empathy.** The fantasy subscale of the Interpersonal Reactivity Index (IRI; [31]) was used as an indicator of trait empathy as it has previously been shown to be related to the degree to which people enjoy sad music [18]. The fantasy subscale taps respondents' tendencies to transpose themselves imaginatively into the feelings and actions of fictitious characters in books, movies, and plays. Possible scores on the fantasy subscale of the IRI range from 7–35. Participants respond on a five-point Likert scale which ranges from 'Does not describe me well' (1) to 'Describes me very well' (5).

**Musical stimuli.** In collaboration with a professional music composer and global music production company, informed by existing research on music preferences in the general population [41–44] and music preferences in pain management contexts [5, 7, 45], the lead researcher (C.H.) co-created a bespoke track for use in the present study, with multi-instrumentalist and composer, Anatole. This track was manipulated to create two versions varying in musical complexity by digitally removing different components of the melody, ornamentation, and percussion. Great care was taken to minimise the difference between the two versions, while achieving a perceivable difference. The core structure of the track was identical in both versions, such that the more complex track had a greater number of elements included. The *high complexity* track was designed to lead to sustained engagement and enjoyment from a general audience across all age groups. Specifically, the composer was asked to create a piece of music with a build-up of tension that gave way to a great sense of release in the last section of the track. This is a framework that is typically used in classical music, and helps to build a sense of anticipation [24] which may lead to higher levels of engagement. Note that the term "high complexity" is used as a description relative to the other condition, indeed both versions were still accessible to a general audience. To make the music more accessible to a general audience, the both versions of the track were kept quite short (3 minutes and 24 seconds). Although a familiar musical structure was used, different instruments and field recordings were introduced at different points to increase the level of novelty and surprise experienced by listeners.

The composer also varied the quality of different musical instruments, creating a slight sense of distortion on several sounds to create a sense of novelty using a familiar sound. A sense of novelty was emphasised during the creation of the track because novelty and surprise in musical experiences are related greater levels of pleasure and neural reward [46, 47]. Percussion was used extensively through the track to create a syncopated rhythm pattern as this tends to be preferred by a general audience [41] and distortion was used to emphasise the crescendo of the track. The tempo increased over the course of the track from 80 beats per minute to 120 beats per minute. The *low complexity* version of the track was much simpler than the main track. This version of the track kept the strings, piano, and bass of the original version but did not have any percussion, harmony, acoustic feedback, or field recordings, and had a consistent tempo of 80 bpm.

## Data analysis

Multilevel modelling facilitates the simultaneous analysis of lower-level variables (e.g. individual effects) and higher level effects (e.g. group effects), and enables one to examine how variables from different levels interact together [48]. An additional benefit of multi-level modelling is that it does not rely on the assumption of independence of the residuals, which means that it can be used where data is clustered which can occur in research designs with independent groups. This was particularly important in the current study because the

Intraclass Correlation Coefficient for each model identified that there was a relatively high degree of interdependence between the residuals. The ICC quantifies the degree of resemblance of the observations belonging to the same cluster and can range from 0 to 1. An ICC of 0 indicates perfect independence of the residuals [48]. In the current study the ICC was 0.67 for pain intensity, and 0.67 for pain unpleasantness, which can be considered as a very high level of homogeneity [49] and underpins the necessity of using a multilevel modelling approach. The details of each multilevel model are presented in the results section. All statistical analyses were conducted on SPSS version 26.

## Results

### Descriptive statistics

Descriptive statistics for the two dependant variables pain intensity and pain unpleasantness were calculated for each of the four experimental conditions (See Table 2). Inspection of the mean scores indicate that the perceived choice condition with high complexity had the greatest mean reductions in pain intensity (-11.04, *SD* 13.91), and pain unpleasantness (-13.67, *SD* 15.46).

### Manipulation checks

Before testing the main hypotheses, a number of pre-requisite checks were performed and are reported first. Specifically before testing the first two hypotheses (H1 and H2), we tested if the tracks were comparable in terms of participants aesthetic and emotional responses towards them (H3).

**Aesthetic responses to high complexity and low complexity music.**   In line with Berlyne's model of aesthetic engagement [50] that positions aesthetic appeal as a combination of optimal complexity with optimal arousal complexity, enjoyment, interest, boredom, and attention ratings were collected to determine the aesthetic appeal of the two tracks. Participants rated both tracks highly in terms of enjoyment, interest, and the degree to which the track captured their attention, and low in terms of boredom, indicating that overall, both tracks were aesthetically pleasing to participants. A manipulation check was completed on participants' aesthetic responses to the two tracks to examine whether music that was considered more complex by the composer was perceived as more complex by the participants. Independent samples t-tests were used to demonstrate that high complexity music was rated as significantly more complex than the low complexity music, $t(558) = -3.41$, $p < .001$ 95% CI [-12.00, -3.22] Cohen's d = -.29. This indicates that participants did perceive the 'high complexity' track as more complex than the 'low complexity' track.

**Emotional ratings of high complexity and low complexity music.**   Participants provided emotional ratings on the GEMS-9 [40] after listening to the music track and completing pain

**Table 2. Pain reduction scores for each experimental condition.**

|  | 1 | 2 | 3 | 4 |
|---|---|---|---|---|
| Experimental Condition | No Choice LC | No Choice HC | Perceived Choice LC | Perceived Choice HC |
| n | 73 | 70 | 67 | 76 |
|  | *M (SD)* | *M (SD)* | *M (SD)* | *M (SD)* |
| Pain Intensity Change | -9.38 (14.01) | -8.59 (12.15) | -8.89 (11.66) | -11.04 (13.91) |
| Pain Unpleasantness Change | -11.80 (16.10) | -10.60 (15.68) | -12.33(13.71) | -13.67(15.46) |

*Notes*. Abbreviations *M* Mean, *SD* Standard Deviation, LC Low Complexity Music, HC High Complexity Music

ratings. High complexity music was rated as statistically significantly lower in terms of Tenderness, $U(286) = 8464.5$, $Z = -2.59$, $p < .01$ and Sadness, $U(286) = 887.5$, $Z = -2.59$, $p < .05$. There were no significant differences in ratings for Wonder, Tension, Activation, Power, Peacefulness, Transcendence or Nostalgia between the *high* complexity and *low complexity* music. Overall, this demonstrates that high complexity music was rated as less sad and less tender.

**Independence of individual measures.** Initially, it was planned to also include individual scores from the Barcelona Music Reward Questionnaire (BMRQ; [28]) and the fantasy subscale of the Interpersonal Reactivity Index (IRI; [31]). However, statistically significant correlations between the BMRQ, IRI and the GMSI meant that this approach would not be suitable as it would lead to covariance in the linear model. Instead only the subscales of the Goldsmiths Musical Sophistication Index were included in the linear modelling.

**Changes in pain scores after music listening.** A key question in this analysis was to identify if the bespoke music tracks could lead to a statistically significant decrease in pain scores. First, we examined if there were any decreases in pain scores overall. Paired sampled t-tests were used to compare pain scores before and after listening to music, for pain intensity and pain unpleasantness.

Overall there was a significant decrease in pain intensity, $t(285) = 12.407$, $p < .001$; 95% CI [8.00, 11.02] Cohen's d = .734, of 9.51 ($SD = 12.97$) after music listening. There was also a significant decrease in pain unpleasantness, $t(285) = 13.39$, $p < .001$ 95% CI [10.30, 13.85] d = .792, of 12.08 ($SD = 15.26$) after music listening. These effect sizes are at the upper range of what was expected based on the previously published meta-analysis which reports an effect size of g = .23 (5), and so we proceeded to test out hypotheses. The distributions of pain scores and their changes in response to music in each of the experimental condition are depicted in violin plots in Fig 2.

## Hypothesis testing

For the core analysis we wanted to test hypothesis 1: that increased perceived control predicts decreases in pain intensity and pain unpleasantness and hypothesis 2: that pain experience will

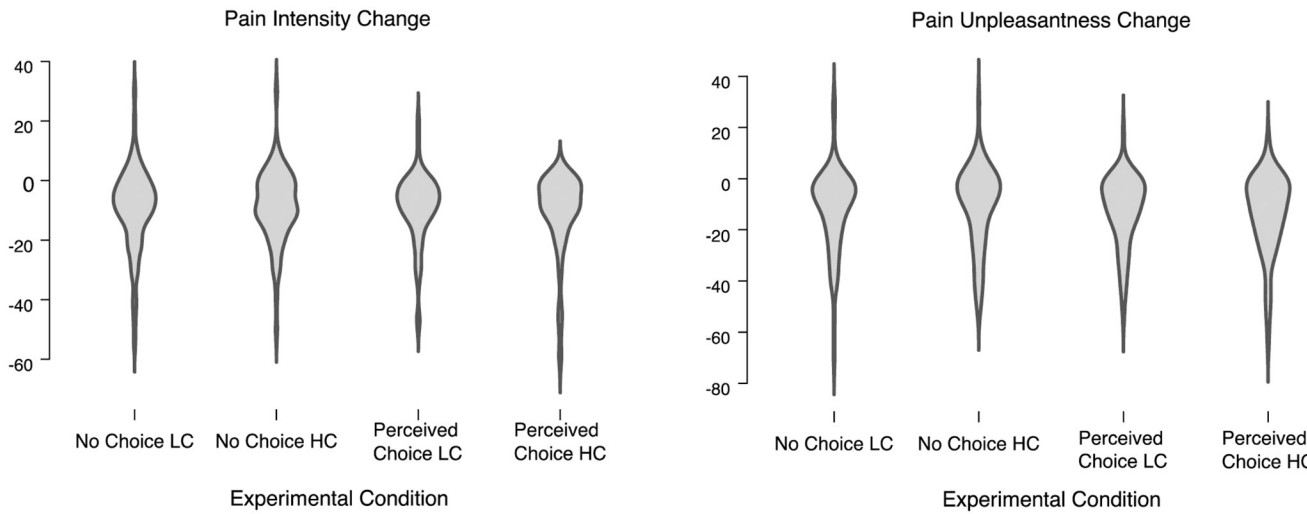

**Fig 2. Violin plots of pain scores in each experimental condition.** Plots depict the distributions of pain scores and their changes in response to music in each of the experimental condition.

be different between the low music complexity and high music complexity conditions. We also recognise that individual attributes related to musicality (specifically active music engagement) may contribute to the analgesic responses to music for acute pain, but we make no specific directional prediction of how this might interact with music choice or music complexity. To address each hypothesis a multilevel model was used to compare the independent variables of level of choice, and music complexity while also accounting for the predictors of Active Engagement and Age. There were two levels for music complexity (high, low) and two levels for choice (perceived choice, no choice). The analysis was conducted for reports of both pain intensity and pain unpleasantness. Multilevel modelling was used to examine the role of individual effects related to musicality (e.g. GMSI Active Engagement Subscale) and higher level effects related to the music group which varied in terms of music complexity (high, low) and music choice (perceived choice, no choice) on the outcomes of pain intensity and pain unpleasantness. We had theoretical reasons to believe that there could be an interaction between different forms of musicality and music complexity, which warranted the investigation of cross-level interactions [39]. Pain reduction scores were calculated by subtracting post-music pain ratings from pre-music pain ratings. A separate linear model was calculated for each of the dependant variables, pain intensity, and pain unpleasantness.

**Pain intensity ratings.** A three-step approach to linear modelling was used [48] and the parameters for the null model and final model are presented in Table 3. As a first step we built an empty model and calculated the ICC. The ICC was 0.67, meaning that 67% of the variance in pain intensity reduction scores could be explained by music group condition (a large

**Table 3. Results of multilevel modelling for pain intensity reductions.**

| Level and Variable | Null Model | | | Final Model | | |
|---|---|---|---|---|---|---|
| | Estimate (SE) | $CI_L$ | $CI_U$ | Estimate (SE) | $CI_L$ | $CI_U$ |
| Level 1 | | | | | | |
| Intercept | -9.51 (9.19) | -49.04 | 30.01 | 7.38 (10.67) | -13.62 | 28.37 |
| Age | | | | -.09(.06) | -.21 | .02 |
| Active Engagement | | | | -.45(.15)** | -.74 | -.15 |
| Level 2 | | | | | | |
| Choice (No Choice vs Perceived Choice) | | | | -14.00(6.72)* | -27.23 | -.77 |
| Track (High vs Low Complexity) | | | | 1.17 (1.54) | -1.87 | 4.20 |
| Cross Level Interaction | | | | | | |
| Choice x Active Engagement | | | | .45(.20)* | .06 | .85 |
| Variance Components | | | | | | |
| Within-group variance | 84.38 | | | 165.24 | | |
| Intercept variance (random) | 83.80 | | | 83.23 | | |
| Slope variance (residuals) | | | | 165.24 | | |
| Intercept-slope variance | | | | -.69 | | |
| Model Information Criteria | | | | | | |
| ICC | 0.67 | | | | | |
| -2 Restricted log likelihood | 2275.09 | | | 2247.68 | | |
| Number of Estimated Parameters | 3 | | | 8 | | |

Note: Values in parentheses are standard errors,

*significant at the $p < .05$ level;

**significant at the $p < .01$ level.

$CI_L$ Lower Confidence Interval, $CI_U$ Upper Confidence Intervals. Multiple comparisons were accounted for in each model using a Bonferroni correction.

within-cluster homogeneity), indicating that multilevel modelling was warranted [51–53]. As a second step, we built intermediate models using the GMSI subscales of musicality, and level of choice, and track complexity and we performed a likelihood-ratio test to see whether estimating the slope residuals improved the fit. The p-value of the Log Liklihood Ratio LR $\chi^2$ (2) was used to determine the best model fit. Age was controlled for because it has been related to music engagement in previous studies [54]. A threshold of $p < 0.20$ for the LR $\chi^2$ (2) was used for each model comparison to determine a significant improvement in model fit. A threshold of $p < 0.20$ for the LR $\chi^2$ (2), was chosen based on previous recommendations [48, 55] to balance between the risk of a type 1 error that may occur with an overly inclusive approach to modelling (e.g. a maximalist approach), and a parsimonious approach to modelling that would tend to *exclude* more parameters. The best fitting model for pain intensity was based on GMSI subscale of Active Engagement, Age, Choice and Music Complexity and had a $p < 0.20$ for the LR $\chi^2$ (2) meaning that estimating the slope residual variance and the covariance terms was warranted. As a third step, we built the final model using active engagement, choice, and the cross-level interaction as predictors, and we observed a significant cross-level interaction between perceived choice and active engagement ß = .45, 95% CI [.06, .85]. A simple slope analysis revealed that there was a relatively large effect for perceived choice on pain intensity decreases ß = -14.00, 95% CI [-27.23, -.77], and a small effect of for the individual trait of Active Engagement whereas the effect for music complexity was null for pain intensity, ß = – 0.45, 95% [-1.87, 4.20]. See Fig 3 for residual plots of each hierarchical model. We interpret this result to mean that perceived choice is related to decreases in pain intensity, and this effect is slightly amplified for those who tend to be more actively engaged with music behaviour on a regular basis irrespective of the relative complexity of the music. Furthermore, the complexity of the music also did not impact pain intensity scores, even when age, and musicality was taken into account.

**Pain unpleasantness.** Again, a three-step approach to linear modelling was used [48] to understand pain unpleasantness scores and the parameters for the null model and final model are presented in Table 4. As a first step we built an empty model and calculated the ICC, which was 0.67. This meant that 67% of the variance in pain intensity reduction scores could be explained by music group condition and indicated that multilevel modelling was warranted to understand the pain unpleasantness scores [51–53]. As a second step, we built intermediate models using the GMSI subscales of musicality, and level of choice, and track complexity and we performed a likelihood-ratio test to see whether estimating the slope residuals improved the fit. The p-value of the Log Liklihood Ratio LR $\chi^2$ (2) was used to determine the best model fit. Age was controlled for because it has been related to music preferences [54]. A threshold of $p < 0.20$ for the LR $\chi^2$ (2) was used for each model comparison to determine a significant improvement in model fit. The best fitting model for pain intensity was based on GMSI subscale of Active Engagement, Age, Choice and Music Complexity and had a $p < 0.20$ for the LR $\chi^2$ (2) meaning that estimating the slope residual variance and the covariance terms was warranted. As a third step, we examined the model fit when interaction terms were included, however, given that they did not significantly improve the model fit, they were not included in the final model. A simple slope analysis revealed that perceived choice had a null effect on pain unpleasantness decreases ß = 1.59, 95% CI [-1.98, 5.15], and a small effect of for the individual trait of Active Engagement ß = -.26, 95% [-.50, -.03]. We interpret this result to mean that perceived choice was not related to decreases in pain unpleasantness, and that individual levels of active engagement were a more reliable predictor of decreases in pain unpleasantness. Additionally, music complexity was not related to decreases in pain unpleasantness even when individual levels of active engagement and age were controlled for. Together these findings

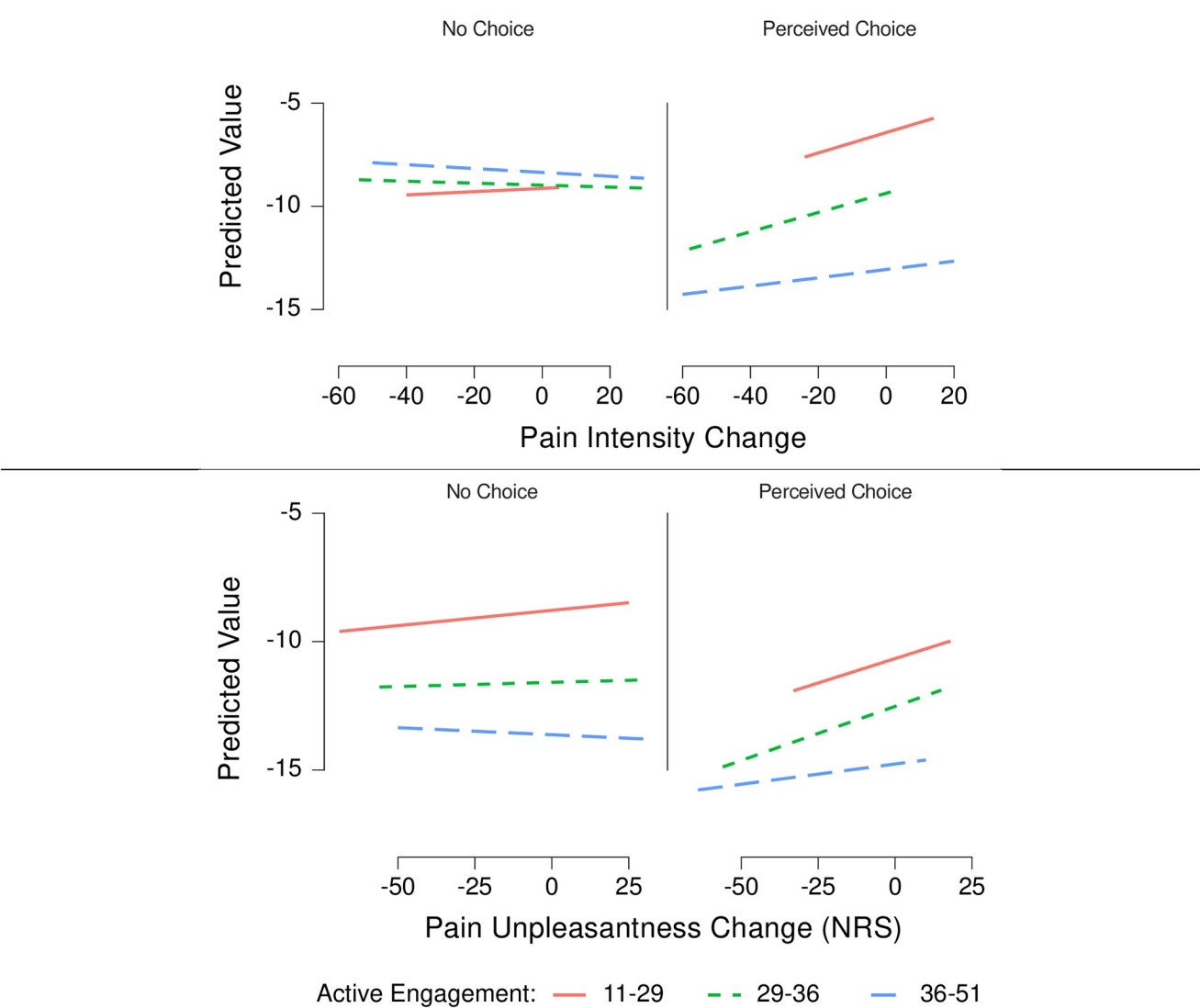

**Fig 3. Residual plots of hierarchical linear models.** Plots depict the relationship between the predicted residual values against the observed values for each model. To depict the choice by active engagement interaction on pain intensity scores plots for each level of choice are shown with model fit depicted according to active engagement. The plot lines fitted to the data are used to illustrate the degree to which the relationship between choice and pain changes depending on different levels of active engagement. The lines illustrate that higher levels of active engagement predict larger decreases in pain intensity in the perceived choice condition. Individual levels of active engagement were also the strongest predictor of decreases in pain unpleasantness.

indicate that the role of cognitive agency and active engagement are more likely to reduce pain intensity ratings compared to pain unpleasantness.

## Discussion

To gain a greater understanding of the analgesic potential of music for acute pain, the present study examined three key factors related to music listening intervention success, cognitive agency, music complexity, and individual levels of trait active engagement. Supporting Hypothesis 1, the current study replicated the analgesic effects of cognitive agency using the perceived choice music paradigm, by demonstrating that the act of choosing music can reduce pain intensity, but not pain unpleasantness. Based on these results, we must consider the role

**Table 4. Results of multilevel modelling for pain unpleasantness reductions.**

| Level and Variable | Model | | | | | |
| --- | --- | --- | --- | --- | --- | --- |
| | Null Model | | | Final Model | | |
| | Estimate (SE) | $CI_L$ | $CI_U$ | Estimate (SE) | $CI_L$ | $CI_U$ |
| Level 1 | | | | | | |
| Intercept | -12.08 (10.81) | -58.57 | 34.42 | -.55 (11.80) | -23.80 | -13.62 |
| Age | | | | -.12 (.07) | -.25 | .02 |
| Active Engagement | | | | -.26 (.12)* | -.50 | -.03 |
| Level 2 | | | | | | |
| Choice (No Choice vs Perceived Choice) | | | | 1.59 (1.81) | -1.98 | 5.15 |
| Track (High vs Low Complexity) | | | | .47 (1.81) | -3.10 | 4.05 |
| Variance Components | | | | | | |
| Within-group variance | 232.74 | | | 230.43 | | |
| Intercept variance (random) | 115.97 | | | 115.64 | | |
| Slope variance (residuals) | | | | 230.43 | | |
| Intercept-slope variance | | | | .00 | | |
| Model Information Criteria | 0.67 | | | | | |
| ICC | 0.67 | | | | | |
| -2 Restricted log likelihood | 2367.69 | | | 2344.22 | | |
| Number of Estimated Parameters | 3 | | | 7 | | |

Note: Values in parentheses are standard errors,

*significant at the p < .05 level;

**significant at the $p$ < .01 level.

$CI_L$ Lower Confidence Interval, $CI_U$ Upper Confidence Intervals. Multiple comparisons were accounted for in each model using a Bonferroni correction.

of the cognitive agency of the individual when they choose a piece of music, appreciating that the choice itself may in fact change the way people engage with the music. As outlined in the Cognitive Vitality Model [11], when people are involved in music choice, they become more actively engaged in the music listening experience, which provides the basis for a greater degree of cognitive and emotional engagement with the piece of music, compared to a passive listening session.

Somewhat deviating from Hypotheses 2 and 3, although overall pain ratings changed from being classed as moderate pain to minor pain, music complexity was not related to decreases in pain intensity or pain unpleasantness. Consistent with previous findings [5, 6], there is no evidence that varying one aspect of the musical experience can account for the wide range of analgesic effects on participants. Instead, pain scores decreased by a similar degree in both the high complexity and low complexity track conditions. Considering that the effect sizes were at the upper end of what was expected based on previous results, and that participants rated both tracks highly in terms of a range of aesthetic responses, the findings demonstrate that there was no difference between the versions of the track used in the current study in terms of their effectiveness in reducing pain. Although participants provided slightly different emotional responses in the two different music conditions, this did not result in differences in pain responses. Aside from leading to less feelings of sadness and tenderness, and greater ratings of complexity the high complex track did not differ from the low complex track in terms of participants aesthetic responses or emotional responses. Without more data exploring this further, the most we can suggest here is that the differences in sadness, tenderness, and complexity were not enough to impact pain intensity or unpleasantness.

A novel aspect of the present research relates to the use of bespoke pieces of music specifically designed for the study. Utilising bespoke music ensured that the tracks were unfamiliar to participants, and also accounted for a potential confound observed in most previous research (i.e., when given a choice, participants often select music that is familiar to them) [16]. Using the perceived choice music paradigm [10], any analgesic effects associated with the act of choosing are isolated from the potential effects of any music features. While this paradigm somewhat controls for familiarity, it is always possible that participants have heard the chosen track. The present study employed a track that was specifically designed for this study and at the time of testing was unavailable for listening elsewhere. Therefore, findings demonstrate that the effects of cognitive agency are strong enough to work even when the music is unfamiliar to participants. But while every effort was made to manipulate complexity while controlling all other aspects of the track, it is possible that the perceived tempo may have also varied due to the added percussion and ornamentation, despite the fact that the core structure of the music was identical in both tracks. This could be examined further in psycho-acoustic experiments.

The present findings also demonstrate that the benefits of cognitive agency are amplified among people who report that they often actively engage with music in their everyday life, as measured by the Active Engagement subscale of the GMSI. Previous studies hypothesised that cognitive agency may lead to greater tendency to actively engage with music [10], and the present results further support the interaction between cognitive agency and active engagement on pain. However, it is important to note that the main effect of cognitive agency, and the interaction between cognitive agency and active engagement were only observed in relation to decreases in pain intensity and not pain unpleasantness.

A strength of the present study arises from the use of a naturalistic setting to explore analgesia from music listening in a real-world sample of adults experiencing acute pain. Previous attempts to examine the cognitive mechanisms of music interventions for pain have been predominantly lab-based [5], sometimes utilising abstract tone sequences that are unlikely to lead to an enjoyable or cognitively absorbing experience [45]. In addition, a recent meta-analysis found that, when examined in a laboratory setting, relative to observations conducted in naturalistic settings, smaller relationships are observed between pain behaviour and self-reported pain intensity, suggesting that naturalistic settings give rise to more accurate and reliable reporting [56]. Finally, measuring pain and music analgesia in a real-world setting increases the likelihood that ratings and observed improvements are representative of participants' daily lives [57]. Given that the present study is more ecologically valid than previous lab-based experiments, findings suggest that the cognitive mechanism of cognitive agency as outlined by the cognitive vitality model [11] can meaningfully reduce pain intensity associated with acute pain in day-to-day living. To summarise, lab-based music analgesia experiments are likely to (i) hinder engagement with music, (ii) fail to capture the relationship between self-reported pain and pain behaviour, and (iii) fail to be representative of individuals' lived experience of pain. The present study accounted for these limitations by enabling people to take part outside the laboratory in a naturalistic setting. Further strengths arise from the use of bespoke music tracks controlling for familiarity, attention checks embedded throughout the experiment, and an emphasis on traits of the individual listener that may impact music engagement.

Due to potential carry over effects from one music condition to another [58], the current authors used a between measures design. This is a clear limitation of the current study because a between measures design does not facilitate close inspection of interindividual differences in pain perception. Yet, with our real-world setting, it would not be possible to eliminate carry over effects and this was deemed a more serious concern to be controlled given our use of a pre-listening baseline. An additional weakness of the current study is that there is no group

with no music as a control. This means that the observed changes can only be attributed to the group conditions, and not directly to the music. Future clinical studies should consider addressing these limitations of the current study. Regardless, the present study extends and advances literature on the role of music listening interventions for pain management. The present findings replicate the effects of both music analgesia and cognitive agency in a naturalistic setting with a novel underexplored pain sample, namely those experiencing acute pain, in addition to demonstrating that the effects of cognitive agency persist even with music that is unfamiliar to participants.

The role of personal agency, and the impact of choosing music on subsequent active engagement with music should be considered in future experimental and clinical studies. A robust body of evidence [59] demonstrates the impact of self-chosen music in pain management contexts, and this study helps to provide evidence in terms of why this happens, which may in turn help to maximise these effects. In line with this, future research could explore means of facilitating sustained engagement with music among people with low levels of Active Engagement perhaps with strategies that can enhance musical engagement such as visual imagery strategies. Music listening interventions that promote personal agency are likely to maximise the analgesic pay-off, and provide individuals with proactive approaches for their own pain management.

## Summary and conclusions

The present study replicated the finding that even the illusion of choice has analgesic benefits. When participants felt that they were controlling the music, they reported greater decreases in pain intensity compared to when they had no control of the music. Although this study did not identify the impact of perceived choice on pain unpleasantness, it did replicate the broad finding that perceived choice is more important than music features. Specifically, in this paper the feature of music complexity was targeted, but it was not found to effect reductions in pain intensity or pain unpleasantness scores, even when individual musicality traits were accounted for in the linear model. Previous papers that have targeted other music features such as tempo [4, 6, 7], energy [5], perceived relaxing properties in the music [6] have not found relationships between basic perceptual properties, and decreases in pain perception. Indeed, several meta-analyses support the view that specific features do not predict reductions in pain [4, 39]. Instead, we argue that it is the way that people engage with music (e.g. focussed listening, self-reflection, meaning-making) that mediates the analgesic benefits of music listening. This argument is supported by the number of studies [5, 41] and meta-analyses [4, 7, 39] that continue to demonstrate that personal choice is the strongest predictor of a successful music intervention for pain. The current study extends previous findings that demonstrate the importance of cognitive agency beyond a laboratory setting to a sample who are experiencing real acute pain. This emphasises the importance of making a choice over music in analgesic settings and suggests that decision-making may play a role in the degree to which people actively engage with the music. Despite the observed strength of the effects of cognitive agency, our findings also demonstrate that they are more pronounced for those more actively engaged with music in their everyday life. This suggests that it may be important to consider the extent to which an individual engages with music when planning music therapy interventions. However even those with low reported levels of Active Engagement reported decreases in pain intensity associated with having perceived control of the music, which underscores the importance of facilitating choice and control in music interventions. The present study has implications for both research and practice, emphasising the importance of attending to individuals' cognitive

agency and engagement, while also identifying means of facilitating and supporting sustained engagement with music throughout music listening interventions.

## Supporting information

**S1 File.**
(PDF)

## Acknowledgments

The authors would like to extend their sincere gratitude to the musician and composer, Anatole, who created the bespoke pieces of music used exclusively for this experiment.

## Author Contributions

**Conceptualization:** Claire Howlin, Alison Stapleton, Brendan Rooney.

**Data curation:** Claire Howlin, Alison Stapleton.

**Formal analysis:** Claire Howlin, Brendan Rooney.

**Funding acquisition:** Claire Howlin.

**Investigation:** Claire Howlin, Alison Stapleton.

**Methodology:** Claire Howlin, Alison Stapleton, Brendan Rooney.

**Project administration:** Claire Howlin, Alison Stapleton.

**Resources:** Claire Howlin, Brendan Rooney.

**Software:** Claire Howlin, Alison Stapleton.

**Supervision:** Claire Howlin.

**Validation:** Claire Howlin, Brendan Rooney.

**Visualization:** Claire Howlin, Alison Stapleton.

**Writing – original draft:** Claire Howlin, Alison Stapleton, Brendan Rooney.

**Writing – review & editing:** Claire Howlin, Alison Stapleton, Brendan Rooney.

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
