## [Decision Letter · Decision Letter 0]

25 Jan 2022

PONE-D-21-33092Analgesic effects of music listening predicted by agency and individual characteristics NOT musical featuresPLOS ONE

Dear Dr. Stapleton,

Thank you for submitting your manuscript to PLOS ONE. After careful consideration, we feel that it has merit but does not fully meet PLOS ONE’s publication criteria as it currently stands. Therefore, we invite you to submit a revised version of the manuscript that addresses the points raised during the review process.

We look forward to receiving your revised manuscript.

Kind regards,

Urs M Nater

Academic Editor

PLOS ONE

Journal Requirements:

Reviewers' comments:

Reviewer's Responses to Questions

**Comments to the Author**

1. Is the manuscript technically sound, and do the data support the conclusions?

Reviewer #1: Partly

Reviewer #2: Partly

2. Has the statistical analysis been performed appropriately and rigorously? 

Reviewer #1: No

Reviewer #2: I Don't Know

3. Have the authors made all data underlying the findings in their manuscript fully available?

Reviewer #1: Yes

Reviewer #2: No

4. Is the manuscript presented in an intelligible fashion and written in standard English?

Reviewer #1: Yes

Reviewer #2: Yes

5. Review Comments to the Author

Reviewer #1: Review

Analgesic effects of music listening predicted by agency and individual characteristics

NOT musical features

The authors investigated the role of perceived control, complexity, and active engagement for the effects of music listening on everyday pain. I appreciate the approach of investigating different underlying mechanisms of analgesic effects of music listening in an experimental online setting. However, there is a range of major and minor issues which should be addressed by the authors:

1. The research gaps should be further specified, based on empirical research, and the novelty and the benefit of the study should be more emphasized in the introduction.

a. Regarding the role of control, the authors mention only one study on pain tolerance – which they actually don’t measure themselves. Other studies did not show that participants’ own choice of music (full control) would be associated with stronger analgesic effects than music chosen by others. A more differentiated view would be necessary.

b. The role of music features is widely under-researched, thus general statements about their weak impact should be avoided. It should be clarified if complexity has already been investigated in the context of analgesic effects of music listening or not. Interpretation (s. abstract, conclusions) should not be generalized on music features as a whole, but only on the investigated feature (complexity, evtl. mixed with tempo, s. 12).

c. The authors highlight active engagement and assume that higher levels of musical sophistication may be associated with stronger pain-reducing effects. However, they don’t explain enough why they assume so and if they assume this only for active engagement or also the other sub-components. This and the research gaps should be further clarified.

2. The hypotheses should target the research gaps and should be specified, mainly hypotheses 2 and 3. In hypothesis 2, the role of complexity should become clear. In hypothesis 3, it should become clear which individual attributes are meant. The introduction suggests that musical sophistication or active engagement are targeted, but more individual attributes are investigated. It is not clear if an interaction between complexity and engagement is expected.

3. In line with 2., the title lets assume that several individual attributes and features are targeted, but the introduction doesn’t focus on several ones. This should be adapted.

4. In line with 2., in the methods section, more parameters are introduced than the theoretical introduction lets assume (e.g. emotional responses and other individual traits than active engagement). Also, it is not always clear how and when variables were assessed. It should become clear which parameters are investigated as primary outcome variables for the investigation of the main hypotheses, and which parameters are additionally investigated and for what reasons, including how and when they were assessed.

5. Apart from the research ethics exemption, it is not specified whether the authors followed any ethical guidelines. A data monitoring section including where and how long data is stored, who has access and how confidentiality is ensured is missing.

6. With regard to the study design, the researchers should clarify why they decided for an experimental online setting on persons with different types of pain instead of a controlled laboratory experiment and/or a pre-defined type of pain.

7. It is unclear why the authors examined a sample size of 286 subjects. Was the group size based on expected effect sizes? Was a power analysis performed? Was it a convenience sample?

8. A control group with no music or an alternative stimulus presentation is missing. Decreases in pain intensity and unpleasantness can therefore not be clearly attributed to the music, but only comparisons between the groups can be interpreted. The general effects (and effect sizes) of music listening should be interpreted with more caution.

9. The sample is very heterogenous. Different types of pain, gender, countries, languages, general medication intake, physical diseases, psychological disorders, body mass index, drug consumption, pregnancy, and music-related profession do not seem to be checked or controlled for. Since these parameters can potentially influence pain perception, the interpretation should be done with much more caution.

10. It should be specified how participants were recruited, why a pain rating of 2 in the NRS, and pain for less than 12 weeks were inclusion criteria, if for any of the points mentioned in 9. was controlled for, and, if measured, how the groups can be described with regard to these points.

11. The instructions regarding the process of music listening should be specified.

12. It is not clear why the composed music conditions differ besides complexity also in tempo. For the investigation of complexity, an additional difference in tempo is not necessary and might limit the interpretability of the results. The reasons for this approach should be explained.

13. The authors write in a table note that one item of the GMSI Active Engagement scale was deleted because it did not load well. Given that the authors mention that the GMSI is validated, this approach is unusual. Changing the scale may affect validity. A citation should be given to support this approach, and validity should in best case be checked before interpreting the results from the new scale. The change should be described in the main text. Instead of citing a book for the validation of the GMSI, the research papers should be cited directly. Regarding fig. 2, it should become clear how the cut-off values for Active Engagement were chosen.

14. More details regarding the multilevel modeling are necessary: Were pain intensity and unpleasantness included in one model or separate models? Was the model fit evaluated with a specific parameter? Were all possible effects included and then non-significant effects deleted, or were effects included step-by-step? Were all possible interaction effects investigated or only specific ones, what were the reasons? It is not clear why age was included as a predictor – its role should be introduced based on research.

15. The results of the different multilevel models (at least null model and final model) should be made available for the reader, and more parameters are necessary. For an example and recommendations on multilevel modeling, see e.g.: Aguinis, H., Gottfredson, R. K., & Culpepper, S. A. (2013). Best-practice recommendations for estimating cross-level interaction effects using multilevel modeling. Journal of Management, 39, 1490-1528.

16. General recommendations for the therapeutic context should not be given. Too little is known about interpersonal differences regarding perception, traits, and (emotional) reactions.

17. Limitations are missing in the discussion section.

18. It should become clear earlier that an online study was conducted.

19. A description of potential dropped out participants and how was dealt with them in the analyses is missing.

20. Block randomization should be described in more detail regarding its settings/criteria.

21. Regarding the manipulation check of complexity, another term than “Aesthetic responses” should be chosen because complexity was rated and not the aesthetic evaluation. It should be specified how complexity was evaluated by the participants.

22. In the discussion, the authors write that repeated-measures experiments would not be possible in an acute pain context. However, repeated measures studies are recommended for pain research due to strong interindividual differences in pain perception. In order to prevent carry-over effects, longer time periods between the measures are necessary.

23. L. 217/218: “Low in boredom” is not equivalent to “aesthetically pleasing”.

24. L. 237 and following lines: Wording of the hypotheses should be consistent.

Reviewer #2: The manuscript presents an experimental online study of music induced analgesia in individuals suffering from acute pain conditions. Overall I think that this is an interesting and important study, yet more detailed information is needed in the manuscript to assess its methodological and scientific merits.

The study was performed online in a group of 286 adults experiencing acute pain from various causes. I feel that the studied group needs to be described in more detail in terms of demographics (gender information is missing), as well as in terms of the pain-related metrics. Were there any participants that were recruited but did not finish the study? Did they differ in terms of individual characteristics from participants that finished the study? Additionally, I think that with this kind of online methodologies, there is a significant possibility of introducing all kinds of biases. The manuscript would benefit from a section of the discussion outlining these. Another problem is the huge heterogeneity of pain conditions (back pain, headache, joint pain, neck pain, period pain etc.). This heterogeneity may be in part the cause of large indivitual variance that has been reported and was not controlled for in the statistical model. I feel that this should also be addressed in the discussion.

Furthermore, I am not sure about the phrasing of the hypotheses 2 and 3:

- "Hypothesis 2: There are analgesic benefits from music specifically designed and composed to

maximise individual engagement." In comparison to? Music not maximising individual engagement? I'm not sure if the study addresses this hypothesis. Perhaps it is a matter of phrasing the hypothesis differently.

- "Hypothesis 3: individual attributes related to musicality predict analgesic responses to music for acute pain" Musicality? From what I understand, it was not tested here. Or did you mean active engagement or sophistication?

Also, the Statistical analysis and Results sections need more clarification:

- What was the rationale behind using a mixed model instead of a regular linear model? Did the authors compare ratings before and after music, independently for each of the experimental conditions? Or did they compare "pain reduction scores" (pre-music pain - post-music pain)? This should be explained more thoroughly.

- Descriptive statistics for pain intensity and unpleasantness are much needed. I would also advise including a plot that visualizes the distributions of pain scores and their changes in response to music in each of the experimental conditions.

- A discussion of statistical power and sample size justification would be appropriate.

- I don't understand why the authors decided to use only the Active Engagement subscale of the GMSI and exclude other GMSI subscales from the models.

- In Table 2, do I understand correctly that for unpleasantness, no interactions were tested? Why?

- I don't think that the conclusion that "The present study also replicated the finding that choice is more important than music features..." is valid. The present study addressed one music feature (complexity) by manipulating it systematically. Other musical features (ie. melody, harmony, timbre, genre, production style, syncopation) were not considered but may influence MIA. This conclusion is also apparent in the title of the paper and I think it gives a wrong impression about what has been tested in this study.

Other considerations:

- L180: What about the running time of the low-complexity track? Was that the same as high complexity? If not, the effects could be attributed to different running times.

- How do the differences in emotional responses to high vs low complexity music influence the results? Emotion is an often discussed factor in MIA. I feel this should be addressed in the discussion.

- The OSF repository cannot be freely accessed, as if it has not been made public. This might be intentional, altough as a reviewer, I would gladly look at the preregistration report.

- I don't think I understand the point in the discussion about repeated measures (L336-340). Why is it a limitation if the present study was not using a repeated manipulation?

- Line 62: Might want to check the reference to Brattico & Pearce, 2013

- L88: "At the same time, this study will explore how individual levels of the trait Active Engagement in the general population relate to the analgesic benefits from music listening."

Why general population? Perhaps this sentence should be phrased differently.

- L200: "linear modelling was used because the intraclass correlation was .34, which indicates the need for multilevel modelling" according to whom? A citation is needed here.

- L233: "This indicates that mean pain ratings went from being classed as moderate pain to minor pain." According to which classification? Can we say that for all participants, or is this just a group mean score? Overall, I'm not sure if this is justified.

- In the methods section, it would be great to see brief descriptions of the subscales of GMSI.

- L263: "Together these findings indicate that the role of cognitive agency and active engagement play a bigger role in reducing pain intensity compared to pain unpleasantness." Please check for grammar.

- L365: "The current study furthers these findings by extending them beyond the laboratory to a sample who are experiencing real acute pain and by demonstrating that these effects remain when the music is completely unfamiliar to participants." Please check grammar.

6. PLOS authors have the option to publish the peer review history of their article (what does this mean?). If published, this will include your full peer review and any attached files.

Reviewer #1: No

Reviewer #2: **Yes: **Krzysztof Basiński

---

## [Author Response · Author response to Decision Letter 0]

16 Mar 2022

Dear Dr. Natar,

We sincerely thank the editor and the reviewer’s for their careful consideration of this manuscript, and would like to extend our appreciation for the time and effort taken to do this. The suggestions and comments from the review team have been most helpful to improve the clarity of reporting. 

After careful consideration, we have decided to revise the paper in light of the recommendations made by the two reviewers. We have attached point by point responses to the issues raised as requested, and we hope you will find these to be satisfactory. 

Additionally we have amended the title to reflect the final results more closely. 

Yours Sincerely, 

Claire Howlin

1. The research gaps should be further specified, based on empirical research, and the novelty and the benefit of the study should be more emphasized in the introduction.

Response: On further consideration, the authors agree with the reviewers that a more nuanced and extended consideration of the literature needs to be included, and each of the points below have been addressed. 

a. Regarding the role of control, the authors mention only one study on pain tolerance – which they actually don’t measure themselves. Other studies did not show that participants’ own choice of music (full control) would be associated with stronger analgesic effects than music chosen by others. A more differentiated view would be necessary.

Response: A number of experimental and clinical studies demonstrate the importance of personal choice, and personal agency in mediating the analgesic benefits of music listening have been included in the introduction. The revised document also clarifies that the aim is to extend the previous findings beyond pain tolerance to explore the pain experience. Line 65 - 69 and 78-79

b. The role of music features is widely under-researched, thus general statements about their weak impact should be avoided. It should be clarified if complexity has already been investigated in the context of analgesic effects of music listening or not. Interpretation (s. abstract, conclusions) should not be generalized on music features as a whole, but only on the investigated feature (complexity, evtl. mixed with tempo, s. 12).

Response: Several studies have tested if basic acoustic properties or if combinations of musical features of music are directly related to the analgesic benefits of music engagement, without much success. The previous version of the document overlooked these and the revised document now elaborates on the previous literature exploring the role of musical features in mediating the analgesic benefits for pain has been included from Line 37 - 46. 

c. The authors highlight active engagement and assume that higher levels of musical sophistication may be associated with stronger pain-reducing effects. However, they don’t explain enough why they assume so and if they assume this only for active engagement or also the other sub-components. This and the research gaps should be further clarified.

Response: In the revised document, we now put forward an argument based on the cognitive vitality model that outlines why higher cognitive processes such as cognitive agency and active cognitive engagement with music should be considered to understand how music can help to reduce pain. Line 48 - 59. Additionally a more explicit link is made between individual characteristics and music engagement. line 96-106.

2. The hypotheses should target the research gaps and should be specified, mainly hypotheses 2 and 3. In hypothesis 2, the role of complexity should become clear. In hypothesis 3, it should become clear which individual attributes are meant. The introduction suggests that musical sophistication or active engagement are targeted, but more individual attributes are investigated. It is not clear if an interaction between complexity and engagement is expected.

Response: The hypotheses have been revised in line with the comment above. Indeed, our pre-registered hypotheses were paraphrased in the introduction (but not the discussion) in the previous document. The revised document includes the elaborated pre-registered hypotheses. Line 114 - 124

3. In line with 2., the title lets assume that several individual attributes and features are targeted, but the introduction doesn’t focus on several ones. This should be adapted.

Response: In the revised manuscript the individual traits are introduced more explicitly in the introduction. The title has also been changed. Line 96-106

4. In line with 2., in the methods section, more parameters are introduced than the theoretical introduction lets assume (e.g. emotional responses and other individual traits than active engagement). Also, it is not always clear how and when variables were assessed. It should become clear which parameters are investigated as primary outcome variables for the investigation of the main hypotheses, and which parameters are additionally investigated and for what reasons, including how and when they were assessed.

Response: All of the individual characteristics are now introduced in the introduction of the revised manuscript. Line 96-106. In addition, the revised “Research Questions and Hypotheses” section now clarifies the rationale for their inclusion and their relative importance in the research. Line 130 - 133.

5. Apart from the research ethics exemption, it is not specified whether the authors followed any ethical guidelines. A data monitoring section including where and how long data is stored, who has access and how confidentiality is ensured is missing.

Response: An extended ethics statement and declaration of data storage procedures has now been included in the revised text. In addition it can be added that the anonymous data were collected using a secure online platform which is compliant with Data Protection regulations and downloaded to a cloud-based university server, as an encrypted zip file protected by a strong password. Line 147 - 152.

6. With regard to the study design, the researchers should clarify why they decided for an experimental online setting on persons with different types of pain instead of a controlled laboratory experiment and/or a pre-defined type of pain.

Response: We previously conducted a lab-based experiment using a cold pressor task to test the role of cognitive agency, musical features and individual attributes in mediating the analgesic benefits of music listening. One of the core aims of this study was to test if the findings that we had observed in the lab would hold in a real world context. Line 78 - 78, and 139 - 142.

7. It is unclear why the authors examined a sample size of 286 subjects. Was the group size based on expected effect sizes? Was a power analysis performed? Was it a convenience sample?

Response: Here we pre-registered our recruitment strategy and the inclusion/exclusion criteria before collecting data. We then followed these steps. The revised document now includes further details in relation to the a priori power calculation, and decision for stopping based on the observed power. Only participants who met the full inclusion criteria were included. Line 153 - 161.

8. A control group with no music or an alternative stimulus presentation is missing. Decreases in pain intensity and unpleasantness can therefore not be clearly attributed to the music, but only comparisons between the groups can be interpreted. The general effects (and effect sizes) of music listening should be interpreted with more caution.

Response: The lack of a control group with no music has been highlighted as a limitation of the study, and references are made in the text at the level of the group rather than directly to the music. Line 45-459

9. The sample is very heterogenous. Different types of pain, gender, countries, languages, general medication intake, physical diseases, psychological disorders, body mass index, drug consumption, pregnancy, and music-related profession do not seem to be checked or controlled for. Since these parameters can potentially influence pain perception, the interpretation should be done with much more caution.

Response: While not all of these variable were controlled for, the use of random assignment to the conditions means that it is extremely unlikely that the groups differ systematically on any of these variables. For these reasons it is typical in experimental design to treat the groups as comparable in these regards. Inferential statistics are used in the analysis to estimate the impact of these “noise” variables and remove them from the estimated models as “error” variance. What remains are the statistically significant effects of the systematically measured variables. It is these variables (choice, complexity and active engagement) upon which we make inferences and claims. 

Nonetheless, several factors were taken into consideration when screening participants for inclusion, including drug consumption, and pregnancy. The revised manuscript includes a more extensive description of the recruitment procedure and inclusion criteria on lines 120 -134. One of the core aims of this study was to replicate a finding that was previously demonstrated in lab setting, to a real world clinical sample. For this reason we felt it was appropriate to include a relatively heterogenous sample to represent different types of acute pain, to increase the generalisability of the findings. 

10. It should be specified how participants were recruited, why a pain rating of 2 in the NRS, and pain for less than 12 weeks were inclusion criteria, if for any of the points mentioned in 9. was controlled for, and, if measured, how the groups can be described with regard to these points.

Response: These inclusion/exclusion criteria were specified in the pre-registration before data were collected. The text has been amended to highlight that a pain rating of 2 was used in the inclusion criteria 'to ensure that they were experiencing at least a mild level of pain before music listening', line 171 and that 'pain extending beyond 12 weeks is often classified as chronic pain (31)' according to the international classification of diseases (ICD-11) line 173.. 

11. The instructions regarding the process of music listening should be specified.

Response: The instruction given to participants in the process of music listening is now included in the revised manuscript. Line 191 - 193.

12. It is not clear why the composed music conditions differ besides complexity also in tempo. For the investigation of complexity, an additional difference in tempo is not necessary and might limit the interpretability of the results. The reasons for this approach should be explained.

Response: The variation in tempo is unavoidable, given that extra percussion was added to the track to increase the complexity, the perceived tempo also increases - even though the core structure is unchanged. To highlight the great care that was taken to minimise the difference between the two versions, while achieving a perceivable difference. The revised manuscript has been amended to reflect this at line 256-259.

13. The authors write in a table note that one item of the GMSI Active Engagement scale was deleted because it did not load well. Given that the authors mention that the GMSI is validated, this approach is unusual. Changing the scale may affect validity. A citation should be given to support this approach, and validity should in best case be checked before interpreting the results from the new scale. The change should be described in the main text. Instead of citing a book for the validation of the GMSI, the research papers should be cited directly. Regarding fig. 2, it should become clear how the cut-off values for Active Engagement were chosen.

Response: Checking the validity of a scale would not be possible based on the data collected. However, construct validity can be inferred from the reliability of the measure. It was indeed this analysis that identified the poor loading of one item. It was removed to as to increase the construct validity of the measure. The plot lines fitted to the data are used to illustrate the degree to which the relationship between choice and pain changes depending on different levels of active engagement. The description for figure 2 has been updated to indicate this more clearly. 

14. More details regarding the multilevel modelling are necessary: Were pain intensity and unpleasantness included in one model or separate models? 

Was the model fit evaluated with a specific parameter? 

Were all possible effects included and then non-significant effects deleted, or were effects included step-by-step? 

Were all possible interaction effects investigated or only specific ones, what were the reasons? It is not clear why age was included as a predictor – its role should be introduced based on research.

Response: The revised manuscript now includes more details about the multilevel modelling approach taken in the data analysis section line 284 - 294..

15. The results of the different multilevel models (at least null model and final model) should be made available for the reader, and more parameters are necessary. For an example and recommendations on multilevel modelling, see e.g.: Aguinis, H., Gottfredson, R. K., & Culpepper, S. A. (2013). Best-practice recommendations for estimating cross-level interaction effects using multilevel modelling. Journal of Management, 39, 1490-1528.

Response: The revised manuscript now reports the results of the multilevel models according to the best practice guidelines recommended by the reviewer, and now includes information about the null model and additional parameters in table 3 and table 4. 

16. General recommendations for the therapeutic context should not be given. Too little is known about interpersonal differences regarding perception, traits, and (emotional) reactions.

Response: General recommendations for therapeutic contexts have been replaced with the relevance of these findings to future clinical and experimental studies has been highlighted. Line 458 - 459.

17. Limitations are missing in the discussion section.

Response: The discussion has been amended to reflect the limitations of the study more clearly from line 447 to 454.

18. It should become clear earlier that an online study was conducted.

Response: The first section of the methods section now indicates that this study was conducted online. "The experiment was presented online using the Qualtrics platform on line 184.

19. A description of potential dropped out participants and how was dealt with them in the analyses is missing.

Response: From an initial sample of 585 participants that completed the survey, 286 people completed all components of the experimental procedure. The revised manuscript includes a more extensive description of the recruitment procedure and inclusion criteria on lines 120 -134.

20. Block randomization should be described in more detail regarding its settings/criteria.

Response: The term Block randomisation was misused in error and the details of simple randomisation to independent groups is now clarified.

‘using the Qualtrics randomiser, participants were randomly allocated to one of the four conditions:’

21. Regarding the manipulation check of complexity, another term than “Aesthetic responses” should be chosen because complexity was rated and not the aesthetic evaluation. It should be specified how complexity was evaluated by the participants.

Response: A number of ratings were collected to determine the aesthetic appeal of the music including complexity, enjoyment, boredom, interest, attention in line with Berlyne's model of aesthetic engagement. The revised manuscript has now been amended to reflect this from line 311 - 315.

22. In the discussion, the authors write that repeated-measures experiments would not be possible in an acute pain context. However, repeated measures studies are recommended for pain research due to strong interindividual differences in pain perception. In order to prevent carry-over effects, longer time periods between the measures are necessary.

Response: The discussion has been amended to reflect more clearly that the between participants design is a limitation of this study on line 452-459.

23. L. 217/218: “Low in boredom” is not equivalent to “aesthetically pleasing”.

Response: A number of ratings were collected to determine the aesthetic appeal of the music including complexity, enjoyment, boredom, interest, attention in line with Berlyne's model of aesthetic engagement. The revised manuscript has been amended to highlight this from line 306 - 310.

24. L. 237 and following lines: Wording of the hypotheses should be consistent.

Response: the wording of the hypotheses has now been amended to be more consistent.

Reviewer #2: The manuscript presents an experimental online study of music induced analgesia in individuals suffering from acute pain conditions. Overall I think that this is an interesting and important study, yet more detailed information is needed in the manuscript to assess its methodological and scientific merits.

The study was performed online in a group of 286 adults experiencing acute pain from various causes. I feel that the studied group needs to be described in more detail in terms of demographics (gender information is missing), as well as in terms of the pain-related metrics. Were there any participants that were recruited but did not finish the study? Did they differ in terms of individual characteristics from participants that finished the study? Additionally, I think that with this kind of online methodologies, there is a significant possibility of introducing all kinds of biases. The manuscript would benefit from a section of the discussion outlining these. Another problem is the huge heterogeneity of pain conditions (back pain, headache, joint pain, neck pain, period pain etc.). This heterogeneity may be in part the cause of large individual variance that has been reported and was not controlled for in the statistical model. I feel that this should also be addressed in the discussion.

Response: The revised manuscript now includes additional details (including gender and pain type) about the overall sample and the composition of each group has been provided in the demographics section and table 1. Additionally more detailed information has been included in relation to our screening and recruitment procedure for the study on Line 120-134. The use of random assignment to the conditions means helps to ensure that the groups do not differ systematically on any of these variables. For these reasons it is typical in experimental design to treat the groups as comparable in these regards. Inferential statistics are used in the analysis to estimate the impact of these “noise” variables and remove them from the estimated models as “error” variance. What remains are the statistically significant effects of the systematically measured variables. It is these variables (choice, complexity and active engagement) upon which we make inferences and claims. One of the core aims of this study was to replicate a finding that was previously demonstrated in lab setting, to a real world clinical sample. For this reason we felt it was appropriate to include a relatively heterogenous sample to represent different types of acute pain, to increase the generalisability of the findings.

Furthermore, I am not sure about the phrasing of the hypotheses 2 and 3:

- "Hypothesis 2: There are analgesic benefits from music specifically designed and composed to

maximise individual engagement." In comparison to? Music not maximising individual engagement? I'm not sure if the study addresses this hypothesis. Perhaps it is a matter of phrasing the hypothesis differently.

- "Hypothesis 3: individual attributes related to musicality predict analgesic responses to music for acute pain" Musicality? From what I understand, it was not tested here. Or did you mean active engagement or sophistication?

Response: The hypotheses have been revised in line with the comment above. Indeed, our pre-registered hypotheses were paraphrased in the introduction (but not the discussion) in the previous document. The revised document includes the elaborated pre-registered hypotheses that clarify the issues raised. Line 114 - 124

Also, the Statistical analysis and Results sections need more clarification:

- What was the rationale behind using a mixed model instead of a regular linear model? Did the authors compare ratings before and after music, independently for each of the experimental conditions? Or did they compare "pain reduction scores" (pre-music pain - post-music pain)? This should be explained more thoroughly.

Response: The revised manuscript now includes more details about the multilevel modelling approach taken, and the rationale for this approach in the data analysis section line 279 - 289. Further clarification has also been included to explain how the pain reduction scores were calculated on line 279 - 280. 

- Descriptive statistics for pain intensity and unpleasantness are much needed. I would also advise including a plot that visualizes the distributions of pain scores and their changes in response to music in each of the experimental conditions.

Response: The revised manuscript now includes descriptive statistics for pain reduction scores in terms of pain intensity and pain unpleasantness on line 296 - 300.. Violin plots have also been included that visualise the distributions of pain scores and their changes in response to music in each of the experimental conditions on line 285-286.

- A discussion of statistical power and sample size justification would be appropriate.

Response: Here we pre-registered our recruitment strategy and the inclusion/exclusion criteria before collecting data. We then followed these steps. The revised document now includes further details in relation to the a priori power calculation, and decision for stopping based on the observed power. Only participants who met the full inclusion criteria were included. Line 153 - 161.

- I don't understand why the authors decided to use only the Active Engagement subscale of the GMSI and exclude other GMSI subscales from the models.

Response: A top-down linear modelling strategy with a loaded matrix structure was used. Items that did not significantly contribute to the best model fit were removed from the model. The active engagement sub-scale was the only sub-scale of the GMSI that significantly contributed to the best fitting model. 

- In Table 2, do I understand correctly that for unpleasantness, no interactions were tested? Why?

Response: A top-down linear modelling strategy with a loaded matrix structure was used. Items that did not significantly contribute to the best model fit were removed from the model. Unpleasantness interactions did not significantly contribute to the best fitting model, and so they were removed from the final model.

- I don't think that the conclusion that "The present study also replicated the finding that choice is more important than music features..." is valid. The present study addressed one music feature (complexity) by manipulating it systematically. Other musical features (ie. melody, harmony, timbre, genre, production style, syncopation) were not considered but may influence MIA. This conclusion is also apparent in the title of the paper and I think it gives a wrong impression about what has been tested in this study.

Response: The conclusion has been amended to contextualise the current findings within a growing consensus that supports the argument that music features are not as important as personal choice. line 475 - 485. Additionally the title has been amended to be more precise.

Other considerations:

- L180: What about the running time of the low-complexity track? Was that the same as high complexity? If not, the effects could be attributed to different running times.

Response: The manuscript has been amended to indicate that " both versions of the track were kept quite short (3 minutes and 24 seconds) on line 260.

- How do the differences in emotional responses to high vs low complexity music influence the results? Emotion is an often discussed factor in MIA. I feel this should be addressed in the discussion.

Response: Although participants provided slightly different emotional responses in the two different music conditions, this did not result in differences in pain responses. The text has been amended to note this on line 404 - 410.

- The OSF repository cannot be freely accessed, as if it has not been made public. This might be intentional, although as a reviewer, I would gladly look at the preregistration report.

Response: The OSF repository is now publicly available at and we invite the reviewers to look at the preregistration report. https://osf.io/egqaz

- I don't think I understand the point in the discussion about repeated measures (L336-340). Why is it a limitation if the present study was not using a repeated manipulation? 

Response: The discussion has been amended to reflect more clearly that the between participants design is a limitation of this study on line 452 - 459.

- Line 62: Might want to check the reference to Brattico & Pearce, 2013

Response: The reference to Brattico & Pearce, 2013 has been updated into the correct format. 

- L88: "At the same time, this study will explore how individual levels of the trait Active Engagement in the general population relate to the analgesic benefits from music listening."

Why general population? Perhaps this sentence should be phrased differently.

Response: This statement has been amended 'At the same time, this study will explore how individual levels of the trait Active Engagement relate to the analgesic benefits from music listening.'

- L200: "linear modelling was used because the intraclass correlation was .34, which indicates the need for multilevel modelling" according to whom? A citation is needed here.

Response: Further clarification has been included in the revised manuscript and an additional references were added to support this approach. Line 284 - 294.

- L233: "This indicates that mean pain ratings went from being classed as moderate pain to minor pain." According to which classification? Can we say that for all participants, or is this just a group mean score? Overall, I'm not sure if this is justified.

Response: Upon recommendation from the reviewers this section has been removed. 

- In the methods section, it would be great to see brief descriptions of the subscales of GMSI.

Response: The revised manuscript includes brief descriptions of the subscales of the GMSI on line 229 - 234.

- L263: "Together these findings indicate that the role of cognitive agency and active engagement play a bigger role in reducing pain intensity compared to pain unpleasantness." Please check for grammar.

Response: The text has been amended "Together these findings indicate that the role of cognitive agency and active engagement are more likely to reduce pain intensity ratings compared to pain unpleasantness." Line 384-386.

- L365: "The current study furthers these findings by extending them beyond the laboratory to a sample who are experiencing real acute pain and by demonstrating that these effects remain when the music is completely unfamiliar to participants." Please check grammar.

Response: The text in the current manuscript now reads 'The current study extends previous findings that demonstrate the importance of cognitive agency beyond a laboratory setting to a sample who are experiencing real acute pain'. Line 485-487.

---

## [Decision Letter · Decision Letter 1]

22 Apr 2022

PONE-D-21-33092R1Tune out pain: agency and active engagement predict pain decreases after music listeningPLOS ONE

Dear Dr. Howlin,

Thank you for submitting your revised manuscript to PLOS ONE. You will see that both reviewers have still a number of concerns that prevent us from reaching a final decision at this point. Therefore, we invite you to submit a revised version of the manuscript that addresses the points raised during the review process.

 Please submit your revised manuscript by Jun 06 2022 11:59PM. If you will need more time than this to complete your revisions, please reply to this message or contact the journal office at plosone@plos.org. Please include the following items when submitting your revised manuscript:A rebuttal letter that responds to each point raised by the academic editor and reviewer(s). You should upload this letter as a separate file labeled 'Response to Reviewers'.A marked-up copy of your manuscript that highlights changes made to the original version. You should upload this as a separate file labeled 'Revised Manuscript with Track Changes'.An unmarked version of your revised paper without tracked changes. You should upload this as a separate file labeled 'Manuscript'.

We look forward to receiving your revised manuscript.

Kind regards,

Urs M Nater

Academic Editor

PLOS ONE

Reviewers' comments:

Reviewer's Responses to Questions

**Comments to the Author**

1. If the authors have adequately addressed your comments raised in a previous round of review and you feel that this manuscript is now acceptable for publication, you may indicate that here to bypass the “Comments to the Author” section, enter your conflict of interest statement in the “Confidential to Editor” section, and submit your "Accept" recommendation.

Reviewer #1: (No Response)

Reviewer #2: (No Response)

2. Is the manuscript technically sound, and do the data support the conclusions?

Reviewer #1: Partly

Reviewer #2: No

3. Has the statistical analysis been performed appropriately and rigorously? 

Reviewer #1: Yes

Reviewer #2: No

4. Have the authors made all data underlying the findings in their manuscript fully available?

Reviewer #1: No

Reviewer #2: No

5. Is the manuscript presented in an intelligible fashion and written in standard English?

Reviewer #1: Yes

Reviewer #2: No

6. Review Comments to the Author

Reviewer #1: Review

Tune out pain: agency and active engagement predict pain decreases after music

listening

This study on the role of perceived control, complexity and active engagement for the effects of music listening on everyday pain has been carefully revised by the authors. I appreciate the effort made by the authors and see that the manuscript has improved a lot in structure and clarity. There are still some issues that should be addressed by the authors.

1. The authors improved the introduction in different aspects. One issue should still be considered in order to strengthen the introduction even more. The authors focus on findings on pain tolerance, and aim to extend these findings on everyday acute pain experience. In the part of research questions and hypotheses it turns out that pain intensity and unpleasantness are investigated as the main outcome variables of interest but these parameters are not mentioned before. I recommend to introduce pain intensity and unpleasantness as well as potential existing or a lack of existing findings on these important outcome variables already in the introduction of the manuscript.

2. The research questions and hypotheses are now clearer. Three issues should still be considered in this section:

2a. The authors write that it would be important to make sure that the music tracks are comparable in terms of “aesthetic and emotional responses” (lines 125-126). However, in the next sentence, the third hypothesis only targets the “aesthetic response”, even though both aesthetic and emotional responses are investigated separately. This should be adapted.

2b. I highly appreciate the mentioned pre-registration of the project and its hypotheses. However, even though the project is specified as public, the downloadable zip-folder is empty in the given link to the webpage of osf.io. This is also important regarding the data availability.

2c. As the authors argue, too low complexity of music can lead to boredom, and too high complexity of the music can lead to irritation or over-stimulation (lines 92-94). It can be assumed that the optimal level of complexity would be between low and high complexity. Still, the authors decided to only investigate low and high complexity which might both not lead to optimal results. It is still interesting to investigate the two rather extreme forms of complexity, but the missing “moderate” complexity condition might be a limitation of the study or eventually one possible explanation for not having found an effect for complexity, and could be discussed later in the manuscript.

3. The methods section is now more elaborated, and important paragraphs have been included. One issue still remains to be considered. The percent values of the distribution of the participants (lines 176-177) across Europe (37.41%) and the United States (36%) don’t sum up to 100%, so the question arises if also other continents are involved or if the percent values should be adapted. Also, the other percent values regarding the different types of pain seem to be not completely correct. These percent values should be re-checked.

4. The results are now clearer and very informative details on multilevel analyses are given. Some remaining issues concern mainly the discussion part. In general, some parts of the hypotheses should be answered in more detail in the discussion part. It could be considered to discuss at first the results regarding H1, then H2, then H3, and then findings on the profile of individual characteristics. The following issues should be considered:

4a. The authors write that H1 would be supported by having replicated the analgesic effects of cognitive agency which demonstrated the benefits of choosing music in a pain context (lines 387-389). H1 targets both pain intensity and pain unpleasantness, but choice only had an effect on pain intensity in the depicted results. This should also become clear in the discussion part by mentioning and discussing that the hypothesized effect was confirmed for pain intensity but not pain unpleasantness.

4b. Similarly, the authors discuss the sig. interaction effect between choice and active engagement (lines 421-427). It should also be mentioned and discussed that this interaction effect was found to be sig. for pain intensity but not pain unpleasantness.

4c. The authors conclude that the findings suggested that the cognitive mechanisms outlined by the cognitive vitality model could meaningfully impact acute pain in day-to-day living (lines 437-439). Keeping in mind that the hypotheses were only partly confirmed (see also 4a and b), this conclusion should be formulated in a more differentiated or cautious way. For the same reason, some of the following conclusions and discussions on the analgesic effects of active engagement and choice should rather target specific aspects of pain than pain in general, also in the part “Summary and Conclusions”.

4d. The authors write that in a real-world setting, it would not be possible to recreate the feeling of acute pain once it has been decreased by music engagement (lines 449-450). Since acute pain can sometimes re-occur after some time, I recommend to change the formulation of the sentence.

4e. Keeping in mind that the hypotheses are only partly confirmed regarding the effect of choice on different pain outcomes, that the eligibility criteria were not so many, that different types of pain were mixed in the study, that a between-subjects design was used and the hypotheses have not yet been investigated in a therapeutic context, it is too early to give the recommendation of letting clients select music themselves in a therapeutic context (lines 461-463).

4f. I highly appreciate that the effort was made to compose new music for the study. It should be considered to discuss in the discussion that the complexity conditions differed in tempo also.

5. The new title lets assume that agency and active engagement predicted pain decreases in the different pain outcomes, which has been found only partly. The title would be improved by formulating it in a way that represents more results, for example by writing “the role of…” instead of “predict”.

6. The authors made clearer in the introduction and the discussion that the results and previous findings can only be interpreted for investigated music features. In the abstract it still says “Overall, findings demonstrated that the illusion of choice has analgesic benefits, and that perceived choice is more important than music features” (lines 28-29). The authors should make clear that this can be concluded only for the “investigated” or “selected” or “specific” music features, or write about “complexity/ (eventually tempo)” instead of “music features” in general.

7. In line 452 it should say that there is “no” group with no music as a control.

8. The manuscript should be re-checked regarding typos and small grammar mistakes (for example l. 103: „emotions“ instead of „emotion’s”, l. 133: “ensure” better than “insure”, l. 238: “Barcelona Musical Reward” can probably be deleted, or should be specified otherwise, l. 348: “.” is missing, l. 356 & 357: “were”).

Reviewer #2: Thank you to the authors for revising the manuscript. While much of the issues were clarified and resolved, I am still very concerned about the logic and the reporting of the statistical analysis. Specifically:

1. I still do not understand the merit of using mixed models in an independent samples design. The authors state that model fit was not improved by including a random factor for participants, "Therefore, random factors were excluded from the final models.". If by random factors authors mean random effects (in R terminology), why even consider them if there are no repeated measures for participants? A regular linear model seems an obvious choice in this experimental design. Using it would also enable the authors to quantify the variance explained by their models using (easily-interpretable) R^2 statistics.

2. What do the numbers presented in Tables 3 and 4 stand for? Are they model estimates for each of the predictors? If so, they should be labelled as such.

3. I do not understand the rationale behind "A top-down linear modelling strategy with a loaded mean structure" (L284). How exactly was the initial model specified? Did the results of the initial model reveal the same significant effects as the final model? If so, why remove the non-significant effects? If not, can the authors explain why? How does this influence the interpretation of the results? Why age was included and not other demographic variables, such as gender? Were all possible interactions explored? If so, why? If not, why? Was the procedure automated? Assuming this approach is similar to step-wise regression procedures, when it was set to stop (that is, which model was considered the most "parsimonious")? Was this decision based on log likelihood or on "removing non-significant effects" (L286)? What was the criterion for "non-significant effect"?

4. Also, I am not sure if this approach is necessary, or even valid here. There are clear hypotheses and a simple experimental design. Including all measured individual difference variables in the model could be beneficial, as it would account for any variance arising from these variables (even if they are "not significant"). This would then enable more robust judgments about the presence (or lack thereof) of main effects and interactions. The authors used what looks like an exploratory approach (although it is hard to tell for certain) that is rarely seen in experimental studies and does not seem to be justified.

5. Finally, the way results are presented is puzzling and unclear. For example, the authors state "significant main effects of both choice, F(1, 269) = 4.82; p < .05, and active engagement, F(1, 272) = 4.21; p < .05.". Why F-test results are given? What are these test actually veryfing? If they were testing the significance of main effects (or, perhaps more accurately, model parameters as this is linear modelling), why not use the t-test? Why do degrees of freedom seem to vary between different parameters of the same model? And crucially, where are the parameter estimates reported?

All these issues may potentially bias the results, therefore it is hard to judge the conclusions of this study without claryfing the statistical issues first.

Other issues:

- The added sections and clarifications contain some grammatical errors that need to be correted. I would suggest a thorough proof-reading of the manuscript.

- Figure 1 appears to be missing.

- Describing the track comparisons in terms of "Hypothesis 3" is a bit misleading, as these comparisons are in fact manipulation checks and not verifications of a hypothesis derived from theory. I understand that this is related to pre-registration, but I would suggest not describing these as "H3" in the text.

- L166-167: "In stage one 2691 answered screening questions which were presented one at a time, based on each previous response along with red herring questions to reduce the likelihood that participants would guess the nature of the study." It is unclear what "2691" refers to.

- The authors indicate that the data from the study will be made publicly available in the OSF, yet the repository is empty as of writing this review.

7. PLOS authors have the option to publish the peer review history of their article (what does this mean?). If published, this will include your full peer review and any attached files.

Reviewer #1: No

Reviewer #2: No

---

## [Author Response · Author response to Decision Letter 1]

26 Apr 2022

Response to reviewers. 

We thank each of the reviewers for their time and attention on this manuscript. In response to the reviewers requests we have added further clarification sand details on several points, supplied further details on the data analysis and have re-organised where all of these details are found, and carefully reviewed and revised the entire manuscript to eliminate typos and minor errors. We hope that you will agree that the manuscript has been further strengthened it terms of clarity and structure, and should be of high interest to the readers of PLoS one. 

Thank you for agreeing and taking the time to complete this additional review. Below you will find a point by point response to each of the queries raised. 

1. The authors improved the introduction in different aspects. One issue should still be considered in order to strengthen the introduction even more. The authors focus on findings on pain tolerance, and aim to extend these findings on everyday acute pain experience. In the part of research questions and hypotheses it turns out that pain intensity and unpleasantness are investigated as the main outcome variables of interest but these parameters are not mentioned before. I recommend to introduce pain intensity and unpleasantness as well as potential existing or a lack of existing findings on these important outcome variables already in the introduction of the manuscript.

Response: The introduction now introduces the concepts of pain unpleasantness and pain intensity, and the rationale for using these measures to measure pain. Line 120 - 127.

2. The research questions and hypotheses are now clearer. Three issues should still be considered in this section:

2a. The authors write that it would be important to make sure that the music tracks are comparable in terms of “aesthetic and emotional responses” (lines 125-126). However, in the next sentence, the third hypothesis only targets the “aesthetic response”, even though both aesthetic and emotional responses are investigated separately. This should be adapted.

Response This section of the text has now been amended (Line 143 - 144).

2b. I highly appreciate the mentioned pre-registration of the project and its hypotheses. However, even though the project is specified as public, the downloadable zip-folder is empty in the given link to the webpage of osf.io. This is also important regarding the data availability.

Response: There seems to be a technical issue with the OSF repository, on our side we can see the review at the link https://osf.io/egqaz. Or on the OSF wiki page, it is under the tab 'registrations'. The project was registered on 26th July 2021. The authors have also attached a pdf version of the protocol registration for ease of access.

2c. As the authors argue, too low complexity of music can lead to boredom, and too high complexity of the music can lead to irritation or over-stimulation (lines 92-94). It can be assumed that the optimal level of complexity would be between low and high complexity. Still, the authors decided to only investigate low and high complexity which might both not lead to optimal results. It is still interesting to investigate the two rather extreme forms of complexity, but the missing “moderate” complexity condition might be a limitation of the study or eventually one possible explanation for not having found an effect for complexity, and could be discussed later in the manuscript.

Response: Optimal complexity is a relative term. We used the terms “high” and “low” to describe the conditions relative to each other. The section on “Musical Stimuli” details our thinking behind the design of these conditions. Working with the music composer, the “high complexity” track was designed to lead to sustained engagement and enjoyment from a general audience across all age groups. While it was high relative to the low condition, we can distinguish it from one that is “too high” to be enjoyable to a general audience. We could rename the conditions as “moderate complexity” vs “simple” but we have retained our naming in the revised manuscript for the sake of simplicity. Nevertheless, tis has been clarified in Lines 279 and 280. 

3. The methods section is now more elaborated, and important paragraphs have been included. One issue still remains to be considered. The percent values of the distribution of the participants (lines 176-177) across Europe (37.41%) and the United States (36%) don’t sum up to 100%, so the question arises if also other continents are involved or if the percent values should be adapted. Also, the other percent values regarding the different types of pain seem to be not completely correct. These percent values should be re-checked.

Response: Additional details of the geographical distribution of participants have been included 194-195. The values representing pain are frequency values rather than percentages so they sum to the total number in the sample. All of the values have been re-checked and one value was found to need correction so this was updated. 

4. The results are now clearer and very informative details on multilevel analyses are given. Some remaining issues concern mainly the discussion part. In general, some parts of the hypotheses should be answered in more detail in the discussion part. It could be considered to discuss at first the results regarding H1, then H2, then H3, and then findings on the profile of individual characteristics. The following issues should be considered:

4a. The authors write that H1 would be supported by having replicated the analgesic effects of cognitive agency which demonstrated the benefits of choosing music in a pain context (lines 387-389). H1 targets both pain intensity and pain unpleasantness, but choice only had an effect on pain intensity in the depicted results. This should also become clear in the discussion part by mentioning and discussing that the hypothesized effect was confirmed for pain intensity but not pain unpleasantness.

Response Line 436-437 have been amended to report the findings with more detail. 

4b. Similarly, the authors discuss the sig. interaction effect between choice and active engagement (lines 421-427). It should also be mentioned and discussed that this interaction effect was found to be sig. for pain intensity but not pain unpleasantness.

Response Line 478-480 have been amended to report the findings with more detail. 

4c. The authors conclude that the findings suggested that the cognitive mechanisms outlined by the cognitive vitality model could meaningfully impact acute pain in day-to-day living (lines 437-439). Keeping in mind that the hypotheses were only partly confirmed (see also 4a and b), this conclusion should be formulated in a more differentiated or cautious way. For the same reason, some of the following conclusions and discussions on the analgesic effects of active engagement and choice should rather target specific aspects of pain than pain in general, also in the part “Summary and Conclusions”.

Response Line 491-492 and the summary and conclusion have been re-formulated in a more differentiated way. 

4d. The authors write that in a real-world setting, it would not be possible to recreate the feeling of acute pain once it has been decreased by music engagement (lines 449-450). Since acute pain can sometimes re-occur after some time, I recommend to change the formulation of the sentence.

Response: Line 503-505 have been reformulated to more precisely express the point. 

4e. Keeping in mind that the hypotheses are only partly confirmed regarding the effect of choice on different pain outcomes, that the eligibility criteria were not so many, that different types of pain were mixed in the study, that a between-subjects design was used and the hypotheses have not yet been investigated in a therapeutic context, it is too early to give the recommendation of letting clients select music themselves in a therapeutic context (lines 461-463).

Response: Lines 514-517 have been revised. 

4f. I highly appreciate that the effort was made to compose new music for the study. It should be considered to discuss in the discussion that the complexity conditions differed in tempo also.

Response The impact of the additional ornamentation and percussion on the perceived tempo has been highlighted from line 468-472.

5. The new title lets assume that agency and active engagement predicted pain decreases in the different pain outcomes, which has been found only partly. The title would be improved by formulating it in a way that represents more results, for example by writing “the role of…” instead of “predict”.

Response: The title has been amended to reflect that the outcomes have been found partly. 

6. The authors made clearer in the introduction and the discussion that the results and previous findings can only be interpreted for investigated music features. In the abstract it still says “Overall, findings demonstrated that the illusion of choice has analgesic benefits, and that perceived choice is more important than music features” (lines 28-29). The authors should make clear that this can be concluded only for the “investigated” or “selected” or “specific” music features, or write about “complexity/ (eventually tempo)” instead of “music features” in general.

Response The abstract has been amended to specifically refer to music complexity.

7. In line 452 it should say that there is “no” group with no music as a control.

Response: Line 505 has been amended. 

8. The manuscript should be re-checked regarding typos and small grammar mistakes (for example l. 103: „emotions“ instead of „emotion’s”, l. 133: “ensure” better than “insure”, l. 238: “Barcelona Musical Reward” can probably be deleted, or should be specified otherwise, l. 348: “.” is missing, l. 356 & 357: “were”).

Response: The document has been carefully revised and corrected for typos. 

Reviewer #2: Thank you to the authors for revising the manuscript. While much of the issues were clarified and resolved, I am still very concerned about the logic and the reporting of the statistical analysis. Specifically:

1. I still do not understand the merit of using mixed models in an independent samples design. The authors state that model fit was not improved by including a random factor for participants, "Therefore, random factors were excluded from the final models.". If by random factors authors mean random effects (in R terminology), why even consider them if there are no repeated measures for participants? A regular linear model seems an obvious choice in this experimental design. Using it would also enable the authors to quantify the variance explained by their models using (easily-interpretable) R^2 statistics.

Response The core reason for using a multilevel model in an independent samples design is to account for the hierarchical structure in the data, which can be empirically identified by the presence of an intraclass correlation coefficient of over 0.2. In the current study an ICC of 0.67 is identified which demonstrates that the residuals are not independent, which precludes the use of a regression analysis based on an ordinary least squares calculation. An additional reason for using a multilevel model in this context is because it allows you to simultaneously examine higher order effects from lower order effects, which means you can disentangle which effect is accounting for more of the variance. The paragraph that highlights the rationale for using multi-level modelling has been further amended to highlight this, and includes further citations to support this approach line 274 - 286.

2. What do the numbers presented in Tables 3 and 4 stand for? Are they model estimates for each of the predictors? If so, they should be labelled as such.

Response Additional labels have been added in table 3 and 4. 

3. I do not understand the rationale behind "A top-down linear modelling strategy with a loaded mean structure" (L284). How exactly was the initial model specified? Did the results of the initial model reveal the same significant effects as the final model? If so, why remove the non-significant effects? If not, can the authors explain why? How does this influence the interpretation of the results? Why age was included and not other demographic variables, such as gender? Were all possible interactions explored? If so, why? If not, why? Was the procedure automated? Assuming this approach is similar to step-wise regression procedures, when it was set to stop (that is, which model was considered the most "parsimonious")? Was this decision based on log likelihood or on "removing non-significant effects" (L286)? What was the criterion for "non-significant effect"?

Response: The reporting of the results has been re-written and re-organised to facilitate a clearer representation of the final results. The rationale for using the linear model is detailed in the data analysis section in the method, and the three step process used for the linear modelling approach has been outlined for each of the two dependant variables, and this is reported in the results section. Additional citations have been provided that outline the rationale for using multilevel modelling with independent samples.

4. Also, I am not sure if this approach is necessary, or even valid here. There are clear hypotheses and a simple experimental design. Including all measured individual difference variables in the model could be beneficial, as it would account for any variance arising from these variables (even if they are "not significant"). This would then enable more robust judgments about the presence (or lack thereof) of main effects and interactions. The authors used what looks like an exploratory approach (although it is hard to tell for certain) that is rarely seen in experimental studies and does not seem to be justified.

Response: The core reason for using a multilevel model in an independent samples design is to account for the hierarchical structure in the data, which can be empirically identified by the presence of an intraclass correlation coefficient of over 0.2. In the current study an ICC of 0.67 is identified which demonstrates that the residuals are not independent, which precludes the use of a regression analysis based on an ordinary least squares calculation. An additional reason for using a multilevel model in this context is because it allows you to simultaneously examine higher order effects from lower order effects, which means you can disentangle which effect is accounting for more of the variance. The paragraph that highlights the rationale for using multi-level modelling in the data analysis section of the method has been further amended to highlight this, and the authors have included further citations to support this approach line 274 - 286.

5. Finally, the way results are presented is puzzling and unclear. For example, the authors state "significant main effects of both choice, F(1, 269) = 4.82; p < .05, and active engagement, F(1, 272) = 4.21; p < .05.". Why F-test results are given? What are these test actually veryfing? If they were testing the significance of main effects (or, perhaps more accurately, model parameters as this is linear modelling), why not use the t-test? Why do degrees of freedom seem to vary between different parameters of the same model? And crucially, where are the parameter estimates reported?

Response: The reporting of the results has been re-written and re-organised to facilitate a clearer representation of the final results. The rationale for using the linear model is detailed in the data analysis section in the method, and the three step process used for the linear modelling approach has been outlined for each of the two dependant variables. F statistics have been removed for the models, and confidence intervals have been included with the parameter estimates. Additional citations have been provided that outline the rationale for using multilevel modelling with independent samples. 

All these issues may potentially bias the results, therefore it is hard to judge the conclusions of this study without claryfing the statistical issues first.

Other issues:

- The added sections and clarifications contain some grammatical errors that need to be correted. I would suggest a thorough proof-reading of the manuscript.

Response The document has been carefully revised and corrected for typos. 

- Figure 1 appears to be missing.

Response Figure 1 is submitted as a separate document and appears at the end of the document. 

- Describing the track comparisons in terms of "Hypothesis 3" is a bit misleading, as these comparisons are in fact manipulation checks and not verifications of a hypothesis derived from theory. I understand that this is related to pre-registration, but I would suggest not describing these as "H3" in the text.

Response The final analysis is no longer described as a 'hypothesis 3' to aid understanding. 

- L166-167: "In stage one 2691 answered screening questions which were presented one at a time, based on each previous response along with red herring questions to reduce the likelihood that participants would guess the nature of the study." It is unclear what "2691" refers to.

Response Line 183 has been amended. 

- The authors indicate that the data from the study will be made publicly available in the OSF, yet the repository is empty as of writing this review.

Response: There seems to be a technical issue with the OSF repository, on our side we can see the review at the link https://osf.io/egqaz . Or on the OSF wiki page, it is under the tab 'registrations'. The project was registered on 26th July 2021. The authors have also attached a pdf version of the protocol registration for ease of access.

---

## [Decision Letter · Decision Letter 2]

2 Jun 2022

PONE-D-21-33092R2Title: Tune out pain: agency and active engagement predict decreases in pain intensity after music listeningPLOS ONE

Dear Dr. Howlin,

Thank you for submitting your manuscript to PLOS ONE. We are very close; you will see that one of the reviewers had some additional suggestions on how to further improve your manuscript. After you have addressed those, there will be no need to go through another round of reviews.

We look forward to receiving your revised manuscript.

Kind regards,

Urs M Nater

Academic Editor

PLOS ONE

Journal Requirements:

Reviewers' comments:

Reviewer's Responses to Questions

**Comments to the Author**

1. If the authors have adequately addressed your comments raised in a previous round of review and you feel that this manuscript is now acceptable for publication, you may indicate that here to bypass the “Comments to the Author” section, enter your conflict of interest statement in the “Confidential to Editor” section, and submit your "Accept" recommendation.

Reviewer #2: (No Response)

2. Is the manuscript technically sound, and do the data support the conclusions?

Reviewer #2: Yes

3. Has the statistical analysis been performed appropriately and rigorously? 

Reviewer #2: Yes

4. Have the authors made all data underlying the findings in their manuscript fully available?

Reviewer #2: Yes

5. Is the manuscript presented in an intelligible fashion and written in standard English?

Reviewer #2: Yes

6. Review Comments to the Author

Reviewer #2: Thanks to the authors for revising the manuscript. The statistical analyses section is now easier to understand and I am grateful that the authors decided to refrain from using the term "mixed model" (even though they refuse to admit it directly!). To clarify: linear mixed modeling is a different statistical technique than multi-level approach to linear modeling that has been performed here. As it is right now, the manuscript presents the results in a clear fashion, although some minor issues remain:

1. "A threshold of p < 0.20 for the LR χ2 (2) was used for each model comparison" Why p < .20? There should be some justification to this. Would a smaller p value lead to different parameter selection? This is important is it may potentially influence the results.

2. What does B stand for in the Results? (L424-425) If it is the model estimate, it would probably be more appropriate to use Beta (ß) instead.

3. L437: "Supporting Hypothesis 1, ...". It seems that this hypothesis was only partly supported, given that there was no significant effect on pain unpleasantness. (L132-133: "Here we predict that increased perceived control predicts decreases in pain intensity and pain unpleasantness (H1)."). The authors should address this in the discussion.

7. PLOS authors have the option to publish the peer review history of their article (what does this mean?). If published, this will include your full peer review and any attached files.

Reviewer #2: No

---

## [Author Response · Author response to Decision Letter 2]

7 Jun 2022

Response to reviewers. 

We thank each of the reviewers, and the editor, for their time and attention on this manuscript. In response to the final comments from reviewers, we have added further clarification in relation to the statistical analysis and discussion. 

Additionally, one of the reviewers is quite right to point out that there is a difference between mixed models, and hierarchical models. They can overlap, but there is a difference, and it is important to get that right. We hope that you will agree that the manuscript has been further strengthened it terms of clarity, and should be of high interest to the readers of PLoS one. 

Below you will find a point by point response to each of the queries raised.

1. "A threshold of p < 0.20 for the LR χ2 (2) was used for each model comparison" Why p < .20? There should be some justification to this. Would a smaller p value lead to different parameter selection? This is important is it may potentially influence the results.

Response: The following text was added to provide a justification for this threshold: 

"A threshold of p < 0.20 for the LR χ² (2), was chosen based on previous recommendations (47,55) to balance between the risk of a type 1 error that may occur with an overly inclusive approach to modelling (e.g. a maximalist approach), and a parsimonious approach to modelling that would tend to exclude more parameters."

2. What does B stand for in the Results? (L424-425) If it is the model estimate, it would probably be more appropriate to use Beta (ß) instead.

Response: ß has been inserted to report the model estimate. 

3. L437: "Supporting Hypothesis 1, ...". It seems that this hypothesis was only partly supported, given that there was no significant effect on pain unpleasantness. (L132-133: "Here we predict that increased perceived control predicts decreases in pain intensity and pain unpleasantness (H1)."). The authors should address this in the discussion.

Response: The revised manuscript includes the following text on line 483- 485:

"However, it is important to note that the main effect of cognitive agency, and the interaction between cognitive agency and active engagement were only observed in relation to decreases in pain intensity and not pain unpleasantness."

---

## [Editor Report · Decision Letter 3]

29 Jun 2022

Title: Tune out pain: agency and active engagement predict decreases in pain intensity after music listening

PONE-D-21-33092R3

Dear Dr. Howlin,

We’re pleased to inform you that your manuscript has been judged scientifically suitable for publication and will be formally accepted for publication once it meets all outstanding technical requirements.

Kind regards,

Urs M Nater

Academic Editor

PLOS ONE
---

## [Editor Report · Acceptance letter]

8 Jul 2022

PONE-D-21-33092R3 

Tune out pain: agency and active engagement predict decreases in pain intensity after music listening 

Dear Dr. Howlin:

I'm pleased to inform you that your manuscript has been deemed suitable for publication in PLOS ONE. Congratulations! Your manuscript is now with our production department. 

Kind regards, 

on behalf of

Dr. Urs M Nater 

Academic Editor

PLOS ONE